# Galanthamine Fails to Reverse P-gp-Mediated Paclitaxel Resistance in Ovarian Cancer Cell Lines

**DOI:** 10.3390/biomedicines13122852

**Published:** 2025-11-21

**Authors:** Nélia Fonseca, Mariana Nunes, Patrícia M. A. Silva, Hassan Bousbaa, Sara Ricardo

**Affiliations:** 1Differentiation and Cancer Group, Institute for Research and Innovation in Health (i3S) of the University of Porto, 4200-135 Porto, Portugal; neliafonseca8@gmail.com; 2Applied Molecular Biosciences Unit (UCIBIO), Toxicologic Pathology Research Laboratory, Institute of Health Sciences (IUCS), Cooperativa de Ensino Superior Politécnico e Universitário (CESPU), 4585-116 Gandra, Portugal; patricia.silva@cespu.pt; 3Associate Laboratory i4HB, Institute for Health and Bioeconomy, University Institute of Health Sciences (IUCS), University Polytechnic Higher Education Cooperative (CESPU), CRL, 4585-116 Gandra, Portugal; 4Oral Pathology and Rehabilitation Research Unit (UNIPRO), Institute of Health Sciences (IUCS), Cooperativa de Ensino Superior Politécnico e Universitário (CESPU), Rua Central de Gandra 1317, 4585-116 Gandra, Portugal; hassan.bousbaa@iucs.cespu.pt

**Keywords:** chemoresistance, drug repurposing, galanthamine, paclitaxel, high-grade serous carcinoma, synergy drug tests

## Abstract

**Background**: Ovarian cancer has the poorest prognosis of all gynecological malignancies, largely due to its chemoresistance, which poses significant treatment challenges. In this context, drug repurposing emerges as an innovative strategy that employs non-cancer treatments to interact with various signaling pathways, enhancing chemotherapy efficacy while minimizing toxicity. This study investigated the cytotoxic effects of galanthamine, currently used as an Alzheimer’s disease, as a potential treatment for high-grade serous carcinoma, both individually and in combination with paclitaxel. **Methods:** The Presto Blue assay, viability marker assessments, immunocytochemical analysis of apoptosis, and a cumulative assay were employed to evaluate the functionality of P-glycoprotein. **Results:** The results indicated that galanthamine did not demonstrate cytotoxic or synergistic effects in either high-grade serous carcinoma cell line tested, suggesting that it is not a viable strategy for overcoming paclitaxel resistance in this context. The immunocytochemistry analysis indicated that galanthamine does not affect the expression of proteins related to cell viability and proliferation and is not associated with chemoresistance. Additionally, functional assays showed that galanthamine treatment did not affect its drug efflux function at the cellular level. **Conclusions:** Overall, the results indicate that galanthamine is unsuitable for reversing paclitaxel resistance despite some literature suggesting its potential interaction with P-glycoprotein.

## 1. Introduction

Ovarian cancer (OC) is the eighth most common gynecological cancer in women, with approximately 314,000 women diagnosed with OC and 207,000 deaths worldwide in 2022 [1]. This disease is often diagnosed in later stages and is considered to have the worst prognosis among gynecological malignancies [2,3]. Cytoreductive surgery followed by platinum- and taxane-based chemotherapy [carboplatin and paclitaxel (PTX)] is the standard treatment [4], being necessary in some cases, the administration of neoadjuvant chemotherapy to reduce the tumor and improve survival [5,6,7]. PTX binds to the β-tubulin subunit at the taxane-binding site within microtubules, stabilizing them and thereby disrupting mitosis and ultimately leading to cell death [8,9,10]. While this mechanism is effective against rapidly dividing cancer cells, it also impacts non-tumor cells that rely on microtubule dynamics for normal function, such as neurons, glial cells, epithelial cells, and hematopoietic cells. In particular, neuronal microtubule disruption can lead to axonal degeneration and consequent peripheral neuropathy [8,11].

P-glycoprotein (P-gp) is an efflux transporter belonging to the ATP-binding cassette transporter family that mediates both intrinsic and extrinsic substances, such as anticancer drugs [12,13]. It plays a crucial role in resistance mechanisms in patients treated with carboplatin and PTX, causing their levels to decline and accumulate outside the cells [9,14,15]. One of the major challenges for chemoresistance is the overexpression of multidrug resistance efflux transporters encoded by the multidrug resistance-1 gene, namely P-gp [16], which is primarily responsible for transporting hydrophobic compounds, such as doxorubicin and PTX [9,17]. Due to chemoresistance, it is difficult to identify effective and nontoxic inhibitors to prevent it [18,19]. Alternatively, one strategy to reverse chemoresistance is the combination of taxane-based compounds with repurposed drugs, more precisely, the use of non-oncological drugs, to inhibit the expression and function of P-gp [13,15,20]. Additionally, new approaches are also promising, such as the use of a nanoparticle system that targets folate receptors specific to each cancer and is composed essentially of hydrophobic compounds that facilitate their delivery [21].

Galanthamine (GAL) is a drug used for the treatment of several neurodegenerative diseases, namely the symptomatic treatment of Alzheimer’s disease at mild, moderate, and advanced-moderate stages, which improves global and cognitive symptoms [22]. Despite GAL not being reported as a substrate of P-gp, when testing its interaction with P-gp, a dimer of GAL was able to modulate its efflux, becoming an inhibitor after decreasing its activity due to the binding sites of the efflux pump [23]. In addition, GAL exerted a neuroprotective effect in SK-N-SH cells, a neuroblastoma cell line, which is susceptible to oxidative stress [24]. 

Specifically, GAL was selected due to its well-characterized pharmacological profile as a reversible acetylcholinesterase inhibitor with a favorable safety record in clinical use for Alzheimer’s disease. Additionally, preliminary studies have suggested that GAL may exert inhibitory effects on P-gp, a key efflux transporter implicated in chemotherapy resistance, including resistance to PTX. This dual profile of neuroprotective properties combined with potential P-gp inhibition made GAL an attractive candidate for drug repurposing aimed at overcoming PTX resistance in high-grade serous carcinoma (HGSC). However, we acknowledge that P-gp inhibition is only one of many mechanisms underlying drug resistance; therefore, future studies should also investigate other P-gp inhibitors and combinations to comprehensively assess strategies for reversing resistance.

Our hypothesis was to investigate the therapeutic potential of GAL in OC treatment by evaluating the cytotoxic effect of GAL and the efficacy of combining PTX and GAL in two chemoresistant HGSC cell lines (OVCAR8 and OVCAR8 PTX R C), as well as its interaction and effect on P-gp, as proposed in the schematic representation in Figure 1.

## 2. Materials and Methods

### 2.1. Cell Lines and Culture Conditions

OVCAR8 and OVCAR8 PTX R C were selected as HGSC cell line models, as described as carboplatin-resistant [25] and carboplatin plus PTX-resistant cell lines [26], respectively. OVCAR8 was derived from the ovarian tumor tissue of a patient with HGSC after receiving a high dose of carboplatin [25] and was generously provided by Doctor Francis Jacob at the Gynecological Cancer Center and Ovarian Cancer Research, Department of Biomedicine, University Hospital Basel, Basel, Switzerland. The OVCAR8 PTX R C cell line was derived from the parental OVCAR8 by continuously exposing it to gradually increasing PTX concentrations (2–74.9 nM) over 3 months [26]. Due to time and resource constraints, the current study primarily focused on P-gp-mediated drug resistance using the resistant variant OVCAR8 PTX R C. To address concerns regarding other resistance mechanisms, we also included the parental OVCAR8 cell line, which is non–P-gp positive, to provide comparative insights. Furthermore, we performed a detailed characterization of the resistance index and P-gp expression levels across multiple cell passages of the resistant lines. These analyses confirmed the stability of the drug-resistant phenotype and P-gp expression throughout the experiments. The HOSE6.3 cell line was chosen as a non-tumoral model; it is a human ovarian epithelial cell line derived from a normal ovary and obtained from a patient with a non-malignant condition [27]. The cell lines were cultured in complete culture medium, namely RPMI-1640 medium, GlutaMAX^TM^ Supplement, HEPES (Thermo Fisher Scientific, Waltham, MA, USA), with 10% (*v*/*v*) inactivated and filtered fetal bovine serum (FBS; Biowest, Nuaillé, France) and 1% (*v*/*v*) penicillin/streptomycin (Thermo Fisher Scientific, Waltham, MA, USA), and incubated at 37 °C in a humidified atmosphere with 5% CO_2_. The cells were routinely authenticated through short tandem repeat profiling and confirmed to be free of mycoplasma. 

### 2.2. Drugs

GAL and PTX were obtained from Selleckchem (Houston, TX, USA), dissolved in dimethyl sulfoxide (DMSO; AppliChem, Barcelona, Spain), and stored at −80 °C following the manufacturer’s instructions. Prior to use, an aliquot was diluted in culture medium to achieve the required concentrations. 

### 2.3. Cell Viability

To evaluate the effect of GAL, either alone or in combination with PTX, a Presto Blue (PB) assay was used to assess the cellular viability of each cell line (OVCAR8, OVCAR8-PTX R C, and HOSE6.3). Briefly, 5 × 10^3^ cells/well of OVCAR8 and OVCAR8 PTX R C and 7.5 × 10^3^ cells/well of HOSE6.3 were seeded into a 96-well plate, incubated at 37 °C in a humidified atmosphere containing 5% CO_2_, and allowed to adhere overnight before drug exposure. Further, cells were exposed to increasing concentrations of GAL (7.81 to 1000 µM) and incubated under the same conditions for 48 h. Then, 50 μL of PrestoBlue^TM^ Cell Viability Reagent 1× (Thermo Fisher Scientific, Waltham, MA, USA) was added, and the mixture was incubated at 37 °C in a humidified atmosphere with 5% CO_2_ for 45 min, protected from light. The fluorescence was measured (560 nm excitation/590 nm emission) using a BioTek Synergy^TM^ 2 multi-mode microplate reader (BioTek, Winooski, VT, USA). This methodology was previously described in [28]. IC_50_ and IC_10_ values were determined from dose–response curves using nonlinear regression analysis with a sigmoidal dose–response model. Curve fitting and calculations were performed using GraphPad Prism 8 (GraphPad Software, Inc., San Diego, CA, USA) according to standard protocols.

### 2.4. Drug Interaction Analysis

Combination studies were conducted using the previously described method, with increasing concentrations of both drugs in a fixed ratio, as recommended by Chou–Talalay [29], in which PTX (drug 1) was combined with GAL (drug 2) in a fixed-dose ratio corresponding to 0.25, 0.5, 1, 2, and 4 times the individual IC_50_ and IC_10_ values, respectively, for 48 h. To measure the drug interactions between PTX (drug 1) and GAL (drug 2), we estimated the synergy score using four different models: Zero Interaction Potency (ZIP), Loewe, Bliss Independence, and Highest Single Agent (HSA) reference models, as implemented in SynergyFinder Plus Software 3.0 (Netphar, Faculty of Medicine, University of Helsinki, Helsinki, Finland) [30]. Each model offers a distinct analytical framework: the Loewe model assumes additivity based on identical modes of action, Bliss independence is suited for drugs with independent mechanisms, the Highest Single Agent (HSA) model compares combination effects to the most effective single agent, and the Zero Interaction Potency (ZIP) model integrates both Loewe and Bliss principles to assess deviations from non-interaction. By applying these complementary models, we aimed to provide a more comprehensive and reliable interpretation of potential synergistic or antagonistic interactions between GAL and PTX. 

The synergy score obtained was interpreted according to a specific range, where the interaction between two drugs can be antagonistic (<−10), additive (−10 to 10), and synergistic (>10) [30]. The cNMF algorithm, implemented in SynergyFinder Plus, was used for estimating outlier measurements [30]. All microscopic images were obtained using a Leica DMi1 inverted phase contrast microscope (Leica Microsystems, Wetzlar, Germany) at 50× magnification.

### 2.5. Immunocytochemistry

OVCAR8 and OVCAR8 PTX R C (2 × 10^5^ cells/well) were seeded into 6-well plates in complete media, incubated at 37 °C in a humidified atmosphere containing 5% CO_2_, and allowed to adhere overnight. After 24 h, cells were exposed to the IC_50_ and IC_10_ values of PTX and GAL, respectively, in single and combined treatment. The IC_50_ values of PTX had been previously determined [26]. The concentrations were selected to ensure acceptable cytotoxic effects and viable cell numbers for marker quantification. 

After 48 h, cells were scraped, washed thrice with ice-cold PBS, and fixed with 10% (*v*/*v*) neutral-buffered formaldehyde (AppliChem GmbH, Darmstadt, Germany) for 1 h at room temperature with gentle agitation. The pellet was re-suspended in liquefied HistoGel ™ (Thermo Fisher Scientific, Waltham, MA, USA), then processed and embedded in paraffin. Each block was sectioned using a microtome, and hematoxylin and eosin staining was performed, as previously described in [26]. The slides were deparaffinized and hydrated, and antigen retrieval was performed by heat-induction (98 °C) using a citrate buffer solution (1:100 at pH 6.0; Thermo Fisher Scientific, Waltham, MA, USA) or trypsin-ethylenediamine tetraacetic acid (1:100; Thermo Fisher Scientific, Waltham, MA, USA) for 40 min. The activity of endogenous peroxidase was blocked using a hydrogen peroxide solution of 3% (*v*/*v*) (Thermo Fisher Scientific, Waltham, MA, USA) for 10 min, and each slide was incubated with specific primary antibodies, namely mesothelin (MSLN; 1:50, SP74, Thermo Fisher Scientific, Waltham, MA, USA), P-gp (1:1200; EPR10364-57, Abcam, Cambridge, UK), Ki67 (1:100; SP6, Thermo Fisher Scientific, Waltham, MA, USA), and cleaved caspase-3 (1:100; D175 5A1E, Cell Signaling Technologies, Danvers, MA, USA), The detection was performed using a secondary antibody conjugated with horseradish peroxidase-labeled polymer (Dako REAL™ EnVision™ Detection System Peroxidase/DAB+, Rabbit/Mouse) for 30 min. Diaminobenzidine was then applied to visualize peroxidase activity following the manufacturer’s guidelines. Hematoxylin staining was employed to highlight cell nuclei. Subsequently, the slides underwent dehydration, clearing and were mounted with a medium suitable for microscopy. The staining pattern—whether nuclear, cytoplasmic, or membranous—and the proportion of stained cells (ranging from 0% to 76–100%) were assessed and confirmed by three independent observers. Scores were resolved by consensus in cases of discrepancy. Immunocytochemical images were obtained using a Leica DM2000 LED brightfield microscope (Leica Microsystems, Wetzlar, Germany) at 400× magnification.

### 2.6. Rhodamine-123 Accumulation

For the evaluation of P-gp activity, 1 × 10^5^ cells/well of OVCAR8 and OVCAR8 PTX R C were seeded into a six-well plate and incubated at 37 °C with 5% CO_2_ and a humidified atmosphere overnight. After 24 h, 1 µM of the fluorescent P-gp substrate Rhodamine-123 (RH-123; Sigma-Aldrich, Saint Louis, MO, USA) was added in the absence or presence of 50 µM and 25 µM of GAL and incubated at 37 °C with 5% CO_2_ humidified atmosphere. After 1 h, cells were washed twice with ice-cold PBS and trypsinized. After centrifugation, the pellet was resuspended in ice-cold PBS for analysis. The inclusion of verapamil (20 µM), a well-established P-gp inhibitor, in all experimental replicates served as a positive control to confirm sensitivity and functional P-gp activity under our experimental conditions. The expected increase in RH-123 accumulation with verapamil validated the assay’s ability to detect P-gp-mediated efflux inhibition, thereby strengthening the interpretation of the negative results observed with GAL. 

The mean fluorescence intensity (MFI) was assessed using a BD Accuri C6™ II flow cytometer (BD Biosciences, San Jose, CA, USA), and the data were analyzed using BD Accuri C6^TM^ Plus Software, version 1.0.27.1 (BD Biosciences, San Jose, CA, USA).

### 2.7. Statistical Analysis

All assays were conducted in triplicate across at least three independent experiments. Data are presented as mean ± standard deviation. Statistical analysis was carried out using GraphPad Prism 8 (GraphPad Software Inc., San Diego, CA, USA), employing either one-way or two-way ANOVA, followed by Šidák’s multiple comparison test.

## 3. Results

### 3.1. Galanthamine Does Not Demonstrate an Effect on Cellular Viability of Chemoresistant High-Grade Serous Carcinoma Cell Lines

Initially, we evaluated the antitumor potential of GAL as a monotherapy in two chemoresistant HGSC cell lines, namely OVCAR8 and OVCAR8 PTX R C cells. Both cell lines were exposed to increasing concentrations of GAL ranging from 7.81 to 1000 µM.

At the maximum range allowed, our results demonstrate that GAL presented low efficacy in reducing the cellular viability of both chemoresistant HGSC cell lines (Figure 2A). Indeed, for OVCAR8 PTX R C, the lowest cellular viability of 87.022 ± 5.307% was reached at 125 µM (Figure 2A). Nevertheless, the most cytotoxic effect on OVCAR8 was observed at 250 µM, resulting in 81.475 ± 8.772% cellular viability (Figure 2A). These results indicate that GAL does not exhibit measurable cytotoxicity against OVCAR8 and OVCAR8 PTX R C cells under the tested conditions, preventing the determination of IC_50_ values within the applied concentration range.

In addition, to evaluate the effects of GAL in normal-like cells, we tested the same increasing concentrations of GAL on a human ovarian epithelial cell line (HOSE6.3). Our results showed that GAL had very low efficacy in reducing the cellular viability of HOSE6.3, with 83.124 ± 7.438% cellular viability at 1000 μM, and did not demonstrate significant differences (Figure 2A). No significant differences were observed between the three cell lines after treatment with increasing concentrations of GAL (Figure 2A). In agreement with the previously mentioned results, no morphological differences were observed in cell size or shape among the cell lines or in the number of cells when exposed to increasing GAL concentrations compared with control (Figure 2B).

Overall, GAL’s cytotoxic effect showed a lack of efficacy in reducing the cellular viability of all three cell lines and failed to obtain IC_50_ values within the considered concentration ranges, indicating a lack of significant anticancer efficacy in chemoresistant HGSC and normal-like cells.

### 3.2. The Combination of Paclitaxel with Galanthamine Does Not Demonstrate Efficacy in Reducing Cellular Viability of Chemoresistant High-Grade Serous Carcinoma Cell Lines

Aside from GAL not demonstrating an antitumor effect, we also tested it in combination with PTX, as this may increase the transport of these drugs into tumor cells, resulting in an enhanced cytotoxic effect. After obtaining the IC_10_ values for GAL, we evaluated the effect of PTX combined with GAL using the combination model previously described [28]. Briefly, OVCAR8, OVCAR8 PTX R C, and HOSE6.3 cells were exposed to increasing concentrations of PTX and GAL, both alone and combined, at a fixed ratio of 0.25, 0.5, 1, 2, and 4 times the individual IC_50_ and IC_10_ values, respectively. Additionally, a morphological evaluation was conducted for each treatment condition.

In OVCAR8 cells, the combination of both drugs resulted in a non-significant increase in effect at all concentrations compared with PTX as a single agent (Figure 3A). Additionally, no significant differences in cellular viability were observed when comparing the effect of GAL alone to the control (Figure 3A). Moreover, in agreement with the previously mentioned results, no morphological changes in cell size or shape were observed in OVCAR8 (Figure 3B). Moreover, no differences were observed in the number of cells exposed to increasing GAL concentrations compared to the control (Figure 3B).

Concerning OVCAR8 PTX R C cells, this combination only produced a significant reduction in cellular viability at 2 times the IC_50_ and IC_10_ values compared to PTX alone (*p* < 0.05) (Figure 4A). However, since it is only statistically significant at 2 times the IC_50_ and IC_10_ values of PTX and GAL, respectively, it becomes unsuitable for clinical use. For the remaining concentrations, this combination does not produce a significant reduction in cellular viability when compared to PTX alone (Figure 4A). Moreover, no significant differences in cellular viability were observed when comparing the effect of GAL alone with the control (Figure 4A). In agreement with the previously mentioned results, no morphological changes in cell size or shape were observed in OVCAR8 PTX R C cells (Figure 4B). Furthermore, we did not observe differences in the number of OVCAR8 PTX R C cells exposed to increasing GAL concentrations compared to the control (Figure 4B).

Regarding HOSE6.3 cells, the combination treatment of PTX with GAL also resulted in a non-significant decrease in cellular viability for all concentrations when compared to PTX as a single agent (Figure 5A). Additionally, our results showed that no significant differences in cellular viability were observed when comparing the effect of GAL alone to the control (Figure 5A). Moreover, in agreement with the previously mentioned results, no evident morphological alterations were observed in HOSE6.3 cells under the tested conditions; however, analyses at higher magnification would be required to confirm potential subtle changes (Figure 5B). Additionally, we did not observe differences in the number of HOSE6.3 cells exposed to increasing GAL concentrations compared to the control (Figure 5B).

Overall, the combinatory effect of PTX with GAL does not demonstrate a significant impact to consider this combination for HGSC treatment.

### 3.3. Combining Paclitaxel with Galanthamine Does Not Demonstrate a Synergistic Effect on Chemoresistant High-Grade Serous Carcinoma Cell Lines

As mentioned earlier, the drug combination aims to achieve a therapeutic effect by leveraging the synergistic effect between the two drugs. This helps to reduce therapeutic doses, consequently minimizing associated side effects and overcoming MDR [31].

To investigate the interaction between PTX and GAL, the synergy score was calculated using SynergyFinder Plus, based on four different methods: ZIP, Loewe, Bliss Independence, and HSA reference models. The synergy score for a drug combination is followed by the average of dose combinations resulting in values higher than 10 (synergism) or less than −10 (antagonism), as observed in the dose regions of heat map and 3D synergy maps, i.e., synergistic (red) and antagonistic (green) [30]. For synergy score values from −10 to 10, the interaction between the two drugs is additive (white) [30].

The four methods differ in their null hypothesis of non-interaction, resulting in different synergy scores for a specific dataset. The ZIP model compares the potency of the drug-response curves of both individual and combined drugs, assuming that between two non-interacting drugs, minimal changes occur in their dose–response curves, which do not affect the effectiveness of each other [30]. The Loewe additivity model reflects the response as both drugs are the same, representing a linear relationship between both drugs corresponding to an additive effect [30]. The Bliss Independence model is an approach that considers the impact of both drugs in combination, assuming their actuation is independent and does not interfere with each other [30]. The HSA method is a reference model that considers the interaction between the combined drugs, enhancing the effect of the HSA by comparing it with the most effective single drug response, rather than considering the additive impact of the combined drugs [30].

Concerning OVCAR8, for all the methods evaluated, combining PTX with GAL results in an additive effect, as all the synergy score values are encompassed between −10 and 10. Specifically, the ZIP, Loewe, Bliss Independence, and HSA models presented synergy scores of −1.88 (*p* = 0.131), 1.28 (*p* = 0.393), −1.87 (*p* = 0.232), and 1.29 (*p* = 0.436), respectively (Figure 6 and Table 1). Moreover, for all the models tested, the most synergic area score (red areas) corresponds to 2.5 nM PTX combined with 100 µM GAL (Figure 6 and Table 1).

For OVCAR8 PTX R C, combining PTX with GAL resulted in an additive effect, as indicated by all the synergy score values, which are encompassed between −10 and 10 for all the methods used. Specifically, the ZIP, Loewe, Bliss Independence, and HSA models presented synergy scores of −0.76 (*p* = 0.637), 3.74 (*p* = 0.048), −0.17 (*p* = 0.929), and 3.60 (*p* = 0.066), respectively (Figure 7 and Table 1). Moreover, for ZIP and HSA models, the most synergic area score (red areas) corresponds to 10 nM PTX combined with 25 µM GAL (Figure 7 and Table 1). On the other hand, for Loewe and Bliss Independence models, the most synergic area score (red areas) corresponds to 10 nM PTX combined with 400 µM GAL (Figure 7 and Table 1).

For HOSE6.3, for all the methods used, the combination of PTX with GAL revealed an additive effect, as all the synergy score values are comprehended between −10 and 10. Specifically, the ZIP, Loewe, Bliss Independence, and HSA models presented synergy scores of −3.25 (*p* = 0.000), 1.23 (*p* = 0.010), −3.49 (*p* = 0.001), and 1.78 (*p* = 0.006), respectively (Figure 8 and Table 1). Moreover, for ZIP and Loewe models, the most synergic area score corresponds to 2.5 nM PTX combined with 400 µM GAL and to 2.5 nM PTX combined with 200 µM GAL, respectively (Figure 8 and Table 1). On the other hand, for Bliss Independence and HSA models, the most synergic area score (red areas) corresponds to 2.5 nM PTX combined with 25 µM GAL (Figure 8 and Table 1).

### 3.4. Galanthamine Treatment Does Not Influence the Phenotype of High-Grade Serous Carcinoma Cell Lines

To complement the cell viability assays, we evaluated the phenotypic changes using specific biomarkers after GAL treatment, both alone and in combination with PTX. Briefly, OVCAR8 and OVCAR8 PTX R C were exposed to PTX and GAL alone and in combination at IC_50_ and IC_10_ values, respectively. For immunocytochemistry, we evaluated five biomarkers: MSLN; chemoresistance-associated markers, assessed by P-gp expression; cell proliferation, evaluated by Ki67 expression; and apoptotic markers, measured by cleaved caspase-3 expression. The staining patterns for each biomarker and the evaluation of the percentage of stained cells are summarized in Figure 9, Figure 10 and Figure 11. To validate the results, several positive tissue samples were used as positive controls for the expression of MSLN, P-gp, Ki67, and cleaved Caspase-3.

First, MSLN expression was assessed. MSLN is a glycoprotein present in the cell membrane of mesothelial cells and is overexpressed in specific cancers, such as OC [32]. High MSLN expression is reportedly associated with poor outcomes, reduced overall survival, tumor dissemination, and increased cellular migration, making it a potential biomarker for evaluating the efficacy of specific treatments [32]. MSLN expression was detected in more than 75% of cells in OVCAR8 for all the treatments (Figure 9 and Figure 10). Regarding the OVCAR8 PTX R C phenotype, both treatments (GAL alone and in combination) demonstrated high expression levels (75–100%). Notably, the staining was predominantly expressed in the cytoplasm, rather than in the cell membrane, as observed in the control (Figure 9 and Figure 11).

In OC, resistance to chemotherapy can result from the overexpression of P-gp, a drug efflux pump that belongs to the ATP-binding cassette transporter family [12,13]. This pump is the principal mechanism for PTX resistance in OC patients, and it has been reported that exposure to increasing concentrations of PTX is responsible for the overexpression of this efflux pump, conferring resistance [26]. The results show that OVCAR8 cells exhibit low or residual P-gp expression after treatment with GAL alone and in combination with PTX (<1%). For PTX alone, increase in expression (6–10%) was observed compared with the control (<1%) (Figure 9 and Figure 10). Regarding OVCAR8 PTX R C, all tested conditions did not significantly alter the expression levels, as all conditions presented high levels of P-gp expression (76–100%) (Figure 9 and Figure 11).

To assess the influence of the treatment on proliferation capacity, Ki67 was used as a marker. Ki-67 is a protein located in the nucleus, where it presents active forms of the cell cycle, and its levels decrease in the final stages of mitosis [33]. In addition, due to its proliferative activity, this marker enables us to demonstrate tumor aggressiveness and the proliferative stage of malignant cells [33]. In both HGSC cell lines, all conditions showed increased Ki-67 expression (76–100%), indicating that more than 75% of the cells were in proliferation (Figure 9). This complemented the previous cellular viability assay, which found that GAL alone or in combination with PTX did not influence the cellular proliferation of the two chemoresistant cell lines (Figure 9, Figure 10 and Figure 11).

Finally, to observe the influence of treatment on apoptotic processes, we used cleaved caspase-3 as a marker. Caspases are cysteine proteases important in signaling pathways leading to apoptosis [34]. One of the pathways involved in the activation of caspase-8 upon ligand binding to the death receptor [34]. In turn, this triggers the activation of caspase-3, 6, and 7, and, consequently, the cleavage of specific cellular proteins is critical to cell survival [34]. In OVCAR8 cells, treatment with PTX alone and in combination with GAL resulted in an increase in expression levels (51–75%) compared to the control (2–5%) and GAL alone (<1%) (Figure 9 and Figure 10). Concerning OVCAR8 PTX R C, cells treated with PTX, either alone or in combination with GAL, exhibited low expression of cleaved caspase-3 (11–25%). When treated with GAL alone, a small increase of 6–10% was observed compared with the control 2–5%), indicating that apoptosis mediated by these signaling pathways does not appear to be triggered (Figure 9 and Figure 11).

### 3.5. Galanthamine Does Not Inhibit the P-Glycoprotein Efflux Pump Function

GAL did not demonstrate any antitumoral effect, either alone or in combination with PTX, nor did it alter the cells’ behavior or phenotype. Still, we also assessed this drug’s interaction with P-gp to provide more detailed insights into the previous results. Using an RH-123 accumulation assay, it was possible to measure intercellular fluorescence after exposure to RH-123, a P-gp fluorescent substrate, to assess the functionality of this efflux pump. The percentage of MIF obtained by flow cytometry is proportional to the intracellular uptake and extracellular efflux of RH-123.

As shown in Figure 12, GAL did not demonstrate an increase in RH-123 accumulation for both chemoresistant cell lines compared with the control (considered 100%). For the parental OVCAR8 cell line, when GAL treatment was applied, no significant differences were observed between 50 µM and 25 µM of GAL (115.40 ± 16.75% and 101.90 ± 13.02%, respectively) (Figure 12). Concerning OVCAR8 PTX R C, there were no significant differences in MIF percentages in the presence of 25 µM of GAL (95.41 ± 11.79%) compared with the higher concentration (89.45 ± 0.22%) (Figure 12).

## 4. Discussion

HGSC is a malignancy diagnosed at later stages that presents the worst prognosis, mainly attributed to chemoresistance [2,3]. Therefore, new therapeutics have been developed to improve treatment effectiveness. Drug repurposing is an innovative strategy for reversing MDR, which has already been tested in combination with conventional chemotherapy to target several signaling pathways and allow treatment with lower toxicity [35]. Several repurposed drugs have shown promising results in OC, namely, metformin [36,37,38], statins [39,40,41], and itraconazole [42,43,44]. MDR represents a significant challenge in the management of HGSC, making it difficult to identify effective and non-toxic inhibitors that specifically target P-gp [18,19]. Therefore, further studies are needed to identify putative P-gp inhibitors. One hypothesis proposes that GAL interacts with P-gp; a dimer of GAL modulates P-gp effux, acting as an inhibitor by decreasing its activity due to the binding sites of the efflux pump [23]. Based on the lack of results corroborating the role of GAL in reversing P-gp-mediated resistance, this study aimed to evaluate the therapeutic effect of GAL in OC cell lines, both alone and in combination with PTX, using PB assays to assess cell viability. Additionally, the interaction of this drug with P-gp was evaluated, as was its expression following treatment. 

The use of three cell lines (OVCAR8, OVCAR8 PTX R C, and HOSE6.3) in this evaluation ensures a specific effect on PTX-resistant cell lines compared with PTX-sensitive cell lines and a normal-like cell line, thereby achieving the safety of the treatment. These cell lines were chosen due to their relevance to HGSC, the most common and aggressive histological subtype, and because the resistant variant (OVCAR8 PTX R C) exhibits a stable and well-defined PTX-resistant phenotype. Although the parental non–P-gp-expressing OVCAR8 cell line provides some contrast, additional non–P-gp-resistant models and clinical samples will be needed to fully understand the spectrum of drug-resistance mechanisms across HGSC. Nonetheless, we recognize that resistance mechanisms can vary across different HGSC cell lines, and the inclusion of additional models—such as those derived from other genetic backgrounds or resistance pathways—would improve the robustness and translational value of the findings. 

Our results suggest that the modulatory effect of GAL on P-gp activity may be cell-type dependent. Since our experiments were conducted in only two HGSC cell models, the observed effects should be interpreted within this biological context. Previous studies have reported variable responses to GAL across different cellular systems, indicating that factors such as the expression level of P-gp, intracellular signaling pathways, and metabolic background may influence its activity. Moreover, previous studies reporting a GAL–P-gp interaction were performed under experimental conditions distinct from ours, including differences in cell type, P-gp expression levels, and assay formats, which may explain the divergent results. In our model, the cell line used exhibits a different P-gp activity profile, potentially accounting for the absence of GAL effects. Although molecular docking and protein-binding assays have suggested possible interactions, these approaches were beyond the scope of the current study; we now acknowledge this limitation and note that such analyses could help clarify the discrepancy between prior predictions and our present findings.

PTX resistance is a complex, multifactorial phenomenon. Beyond P-gp–mediated drug efflux, resistance can arise from alterations in microtubule dynamics or tubulin isotype expression, evasion of apoptosis through dysregulation of pro- and anti-apoptotic proteins, and changes in drug uptake or metabolism [45]. Because PTX targets microtubules to disrupt cell division, mutations or modifications in tubulin can reduce drug binding and efficacy. Dysregulation of apoptotic pathways allows cancer cells to evade cell death, for example, through overexpression of anti-apoptotic proteins like Bcl-2 or downregulation of pro-apoptotic factors, thereby diminishing PTX-induced cytotoxicity. Altered drug metabolism, including enhanced detoxification or reduced activation, can also lower intracellular drug accumulation. Signaling pathways such as PI3K/AKT, MAPK, and NF-κB have been implicated in PTX resistance [45], further illustrating its complexity. These multiple mechanisms highlight why single-agent strategies, like GAL, may be insufficient and why combinatorial or multi-targeted approaches are likely needed. While our study focused on P-gp due to its central role in PTX transport and multidrug resistance, GAL may also affect other resistance pathways. Future studies should explore whether GAL modulates additional mechanisms or interacts synergistically or antagonistically with other P-gp inhibitors. 

GAL treatment as monotherapy demonstrated that this compound was ineffective at decreasing the viability of both HGSC cell lines, with no significant differences compared with the control. In normal-like cells, GAL also showed low effectiveness in terms of cellular viability, demonstrating a safety profile for using this treatment in non-tumor cells. The combination of GAL with PTX did not exhibit a cytotoxic effect on the OVCAR8 cell line, and this effect was dose-independent, as high doses did not alter the cytotoxicity or cellular morphology. We found that when the treatment was applied at two times the IC_50_ and IC_10_ values of PTX and GAL, respectively, there was a significant effect on OVCAR8 PTX R C. The IC_10_ concentration of GAL was chosen based on previous pharmacological and *in vitro* studies using similar doses to investigate potential off-target effects. Although this concentration exceeds therapeutic plasma levels, such supraphysiological concentrations are commonly used *in vitro* to probe mechanistic effects, detect possible cytotoxic or modulatory activity, and inform future preclinical strategies. This approach allowed us to assess GAL-mediated modulation under conditions relevant to P-gp activity.

Although certain concentration pairs exhibited synergistic interactions *in vitro*, these interactions did not translate into significant functional synergy in biological assays. This discrepancy suggests that additional cellular factors, such as drug transport, intracellular metabolism, and signaling pathway regulation, may influence the observed efficacy and limit direct interpretation of chemical synergy. However, due to the high concentration, this effect is not practical as the goal is to enhance established treatments without increasing their toxicity. We did not observe significant differences in cell viability and morphology among the remaining concentrations. In HOSE6.3 cells, which are a non-tumoral cell line, the combination did not affect cellular viability or cause morphological differences. Although no overt morphological changes were observed at 50× magnification, we recognize that this resolution may not capture finer cytotoxic alterations such as nuclear condensation, membrane blebbing, or cytoplasmic vacuolization. Future studies should incorporate higher magnification imaging and, where possible, quantitative image analysis tools (e.g., automated cell morphology profiling or high-content imaging) to detect subtle phenotypic changes that may indicate early or mild cytotoxic effects.

Several studies have reported the cytoprotective effects of GAL. According to Kihara et al., GAL was able to protect against an Amyloid-Beta peptide cytotoxicity ranging from 25 µM to 1000 µM [46]. Moreover, in the Mladenova et al. study, it was also observed that its effect on cell death was reduced when administered at concentrations between 100 and 500 µM as a pre-treatment for Amyloid-Beta peptide in neuroblastoma cells [47]. Arias et al. study, GAL treatment at 300 nM before and during thapsigargin treatment resulted in a 60% reduction in the presence of apoptotic nuclei in neuroblastoma and chromaffin cells. Additionally, it exhibited cytoprotection by increasing cellular viability in both cell types [48]. Furthermore, Mortazavian et al. demonstrated that treatment with 1, 10, and 100 μM of GAL increased the cellular viability of human umbilical vein endothelial cells after H_2_O_2_-induced apoptosis, which is responsible for the decrease in cellular viability [49]. In contrast, GAL hydrobromide demonstrated some growth-inhibitory activity and cytotoxicity toward HeLa cells (a human cervical adenocarcinoma cell line) with an IC_50_ of 30 µM, compared with doxorubicin (3.1 µM) [50]. Likewise, GAL induced lower antiproliferative activity than standard treatment [50]. This effect was also observed in our results, where the combined therapy at 200 µM of GAL (twice the IC_10_ value) resulted in significant differences in the OVCAR8 PTX R C cell line. Overall, in all studies presented, GAL demonstrated either a cytotoxic effect or a potentiator of cellular viability, which is in contrast to our results, as they generally showed no influence on cell viability.

The synergistic effects of PTX and GAL were analyzed using four methods: ZIP, Loewe, Bliss Independence, and HSA. Their analysis demonstrated an additive effect in the tested cell lines. In OVCAR8, all four models presented the most synergic area scores at 2.5 nM PTX when combined with 100 µM GAL. For OVCAR8 PTX R C, the model ZIP and HSA present the most synergic area score at 10 nM PTX combined with 25 µM GAL, and the Loewe and Bliss Independence models at 10 nM PTX combined with 400 µM GAL. Lastly, in normal-like cells, the ZIP and Loewe models obtained the maximal scores at 2.5 nM PTX combined with 400 µM GAL and 2.5 nM PTX combined with 200 µM GAL, respectively. The Bliss Independence and HSA models achieved their maximal scores at 2.5 nM PTX combined with 25 µM GAL. While the observed reductions in cell viability reached statistical significance at certain concentrations, these effects did not translate into meaningful biological responses when considering effect size, reproducibility across biological replicates, or potential therapeutic windows. Given the modest magnitude and lack of consistency, we concluded that these findings were unlikely to reflect clinically actionable synergy and thus did not warrant further pharmacodynamic modeling within the scope of this study. 

To the best of our knowledge, combinations of GAL with carboplatin, PARP inhibitors, or bevacizumab have not been studied or described as a possible treatment for HGSC. However, several studies have evaluated GAL as a neuroprotective agent in the context of chemotherapy-induced tumor side effects. For instance, Ezoulin et al. demonstrated that GAL exerted a neuroprotective effect at 300 nM in an oxidative stress-induced neuroblastoma cell line; however, they did not report this effect in normal-like cells to demonstrate its safety profile [24]. Moreover, the combination of melatonin and GAL showed synergetic protection in a neuroblastoma cell line [51]. For *in vivo* models using rodents, Alsikhan et al. and Aldubaya et al. demonstrated that GAL induced a protective effect against the neurotoxicity induced by doxorubicin, reducing inflammatory markers in the brain [52]. Taken together, these results led us to conclude that GAL could be a promising complement to chemotherapy, not because of its effect on tumor cells, but because it can act as a neuroprotective agent against several side effects of chemotherapy that lead to chemobrain and deteriorate the patient’s quality of life.

To better clarify the lack of effectiveness of GAL treatment in combination with PTX in resistant cell lines and specifically understand the mechanisms behind this interaction, we performed an RH-123 accumulation assay. We evaluated cellular phenotype and behavior variations using immunocytochemistry, assessing P-gp expression and the expression of specific OC biomarkers that target cells in proliferation (Ki-67) and cells in apoptosis (cleaved caspase-3). 

MSLN is a biomarker associated with tumor dissemination, cellular migration, and proliferation, and it can be used to evaluate the effect of GAL treatment in OC cell lines [32,53]. Our results showed that MSLN expression was not altered with GAL treatment. However, a shift from membrane to cytoplasmic localization, which potentially reflects alterations in MSLN may indicate changes in protein trafficking, cellular stress responses, or alterations in membrane integrity associated with drug exposure. Although we did not perform quantitative image analysis or co-localization studies to confirm these hypotheses, the observation warrants further investigation, particularly given MSLN’s role in cell adhesion, signaling, and its potential as a therapeutic target in OC. Several mechanisms could underlie its observed distribution. These include post-translational modifications such as glycosylation, proteolytic cleavage generating soluble forms, intracellular trafficking and recycling, interactions with membrane-associated proteins, and modulation by the tumor microenvironment. Future studies could investigate these pathways to clarify the regulation of MSLN localization.

The other resistance-associated biomarker, P-gp, mainly associated with PTX resistance, showed no change in P-gp expression, corroborating the viability assay results. Despite the structural presence of P-gp in the cell membrane, we further explored its function using a functional assay. The results demonstrated that P-gp is present in tumor cells and is functionally active, exhibiting the drug efflux function that is the mechanism behind drug resistance of OVCAR8 PTX R C cell line.

We employed histology-based techniques to assess cellular viability and phenotype kinetics in cell cultures previously established by our group [54,55]. Using immunocytochemistry, Ki67 and cleaved caspase-3 expression were assessed to identify and evaluate the percentage of cells in proliferation and apoptosis, respectively. Through this approach, we found that more than 75% of the cells were in proliferation, indicating that GAL alone or in combination with PTX did not affect the proliferation capacity of either cell line. In OVCAR8, the combined treatment increased expression levels compared with GAL alone and the control. Regarding OVCAR8 PTX R C, combining GAL and PTX resulted in decreased expression levels, which complements the previous results for GAL alone. Although Ki67 is a commonly used marker of proliferation, it is expressed during all active phases of the cell cycle (G1, S, G2, and M) and does not distinguish between specific cell cycle transitions. As such, its consistent expression across treatment conditions may mask more subtle shifts in cell cycle distribution. Future studies employing flow cytometry-based cell cycle profiling would help clarify whether GAL or PTX induces cell cycle arrest at specific phases, thereby providing more mechanistic insight into their effects. Despite the differences in expression levels, GAL did not influence the pathway responsible for cell apoptosis. The reduced activation of cleaved caspase-3 observed in the PTX-resistant cells, even following treatment, suggests that these cells may evade apoptosis through alternative mechanisms. Potential contributors include the overexpression of anti-apoptotic proteins (e.g., Bcl-2, Bcl-xL), downregulation of pro-apoptotic factors (e.g., Bax, Bad), or activation of caspase-independent cell death pathways such as autophagy or necroptosis [45]. Understanding which of these pathways are predominant could inform the design of targeted combination therapies to restore apoptotic sensitivity in resistant OC cells.

The biomarker analysis presented here corroborates the results of previous cellular viability assays. To the best of our knowledge, no studies have tested the expression of these four markers in the context of OC after GAL treatment, as this compound is mainly tested in association with neurological disorders and cancers and is associated with neuroprotective effects instead of cytotoxic effects.

A dimerized GAL was reported to interact with and block P-gp at an IC_50_ value of approximately 0.5 µM [23]. Following this information and based on the previous results, we complement our understanding by testing the interaction of this drug with the function of P-gp. Interestingly, our results demonstrated that GAL binds to P-gp but cannot inhibit it or reverse PTX resistance. In addition, the literature indicates that GAL is not a substrate for P-gp, and its pharmacokinetic analysis suggests that its permeability enables its pharmacological effect [23,56]. In addition, Eriksson et al. demonstrated that GAL exhibits weak or no interaction with P-gp, which agrees with our results [22]. In this case, we observed binding to the pump, and due to its weak interaction, it likely facilitated their efflux, as previously described. To the best of our knowledge, this additional assay represents an innovative approach as it complements the previously obtained cell viability results. This approach provides more detailed information about the cellular morphology, marker expression, and associated compartmentalization. Moreover, it provides further information about the drug’s interaction with the structure that confers MDR.

This study provides a comprehensive, methodologically rigorous evaluation of the potential role of GAL in reversing PTX resistance in HGSC, contributing valuable insights despite the negative findings. Among its strengths are the use of well-established *in vitro* models of PTX resistance and the thorough assessment of drug efficacy, which enhances the reliability of the results. Furthermore, the detailed experimental design and careful control of variables reduce the likelihood of confounding factors influencing the outcomes. However, several limitations should be considered. Firstly, the study is restricted to *in vitro* models, which may not fully capture the complex tumor microenvironment and pharmacokinetics encountered *in vivo.* Therefore, the translational relevance of the findings warrants cautious interpretation. Secondly, the study focuses solely on GAL and does not explore potential synergistic effects with other agents or different dosing regimens, which could provide additional insights. Lastly, the molecular mechanisms underlying PTX resistance are multifactorial, and targeting a single pathway might be insufficient to overcome resistance. Future studies incorporating *in vivo* models and broader combinatorial approaches could further clarify GAL’s potential role in OC therapy.

Given GAL’s well-established neuroprotective effects through cholinergic modulation and its potential influence on oxidative stress and inflammation, future studies using co-culture models of OC and neuronal cells could provide valuable insight into its dual role. Such systems would allow simultaneous assessment of GAL’s impact on tumor cell chemosensitivity and neuronal viability, better reflecting clinical scenarios where neurotoxicity is a limiting factor in chemotherapy. Beyond the direct effects observed in this study, GAL shows potential as an adjuvant in chemotherapeutic regimens due to its low toxicity in normal cells, suggesting a possible neuroprotective role. Combinatorial strategies involving GAL with microtubule-modulating agents or apoptosis agonists could, in theory, enhance paclitaxel efficacy and help overcome multidrug resistance mechanisms. Future studies are warranted to evaluate the safety and clinical effectiveness of such combinatorial approaches. 

In summary, we conclude that the therapeutic effect of GAL does not influence tumoral cells, whether used as a monotherapy or in combination with PTX. We employed an innovative method to corroborate cell viability assays, combining them with a histology-based approach that enabled morphological evaluation and the expression analysis of specific biomarkers to assess viability (cleaved caspase-3) and biomarkers associated with the normal phenotype of the cell lines (MSLN, P-gp, and Ki-67). Finally, we further investigated the activity of P-gp to elucidate the mechanism underlying the results obtained using GAL treatment.

## 5. Conclusions

It is crucial to develop strategies to overcome chemoresistance in patients with HGSC to improve survival. Therefore, it is important to thoroughly evaluate all potential therapeutic approaches in laboratory settings to identify specific drugs that can be used either independently or in combination with existing chemotherapeutic agents to enhance the overall effectiveness of treatment.

To address this critical issue in HGSC management, we assessed the therapeutic potential of GAL as a combined treatment. We found that GAL has no cytotoxic or synergetic effect on the HGSC cell lines used, as well as differences in the expression of specific biomarkers or the function of P-gp. Our results demonstrated that GAL is not a good candidate for reversing PTX-resistance in the HGSC setting. Still, more studies are needed in other HGSC cellular models to corroborate the obtained results. Nevertheless, this study established a robust method using several cellular assays that can be used in the future to test the cytotoxic effects of other drug candidates against HGSCC chemoresistance.

## Figures and Tables

**Figure 1 biomedicines-13-02852-f001:**
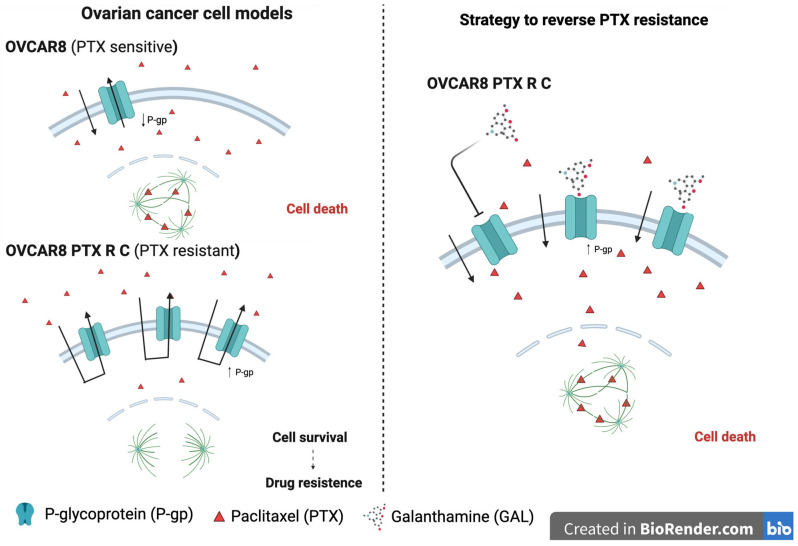
**Schematic representation of a strategy to reverse paclitaxel resistance in high-grade serous carcinoma.** The first representation refers to OVCAR8, a cell line characterized by its sensitivity to PTX, which allows the drug to remain inside the cell and cause cell death. OVCAR8 PTX R C is characterized by P-gp overexpression, leading to PTX efflux and consequent drug resistance. On the right side is proposed a strategy where GAL inhibits P-gp, allowing PTX to remain inside the cell, resulting in cell death. Figure created in BioRender.com. Nélia Fonseca (2023). https://app.biorender.com/illustrations/canvas-beta/656f53fcecb5782c15b13ee6 (accessed on 10 November 2023).

**Figure 2 biomedicines-13-02852-f002:**
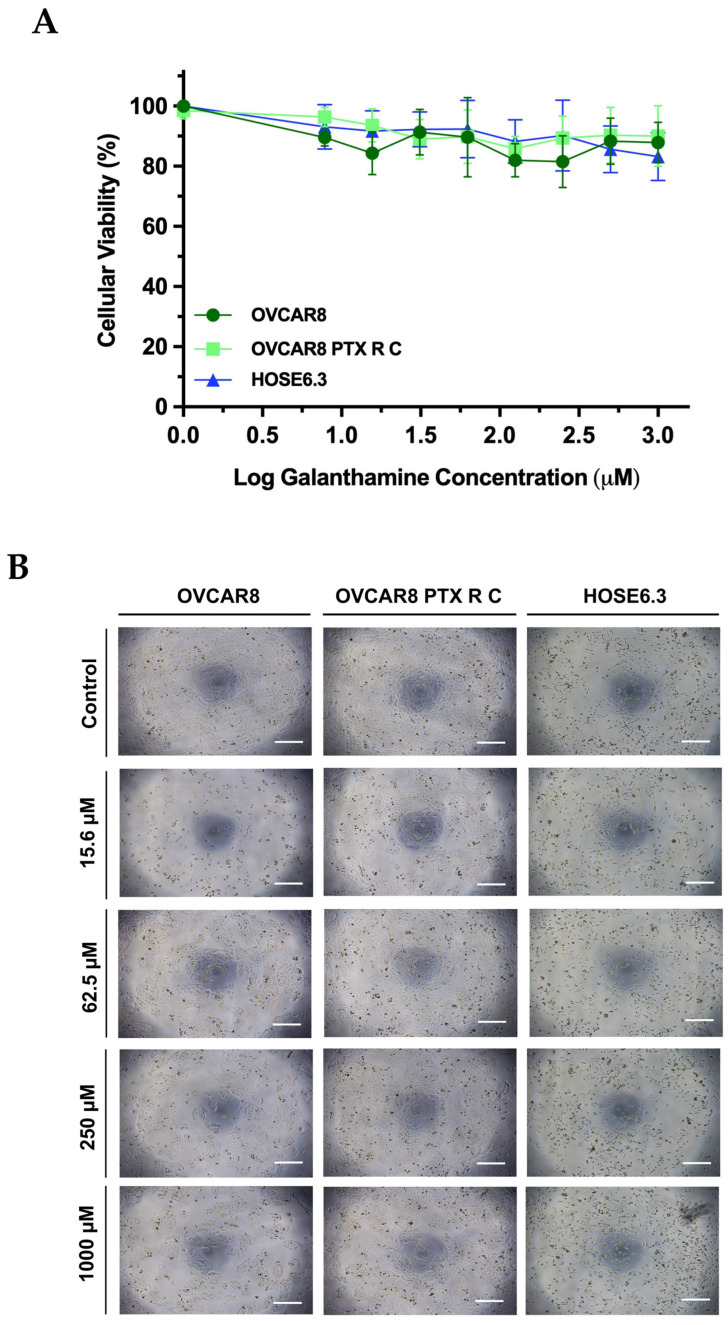
**Galanthamine has not demonstrated an effect on the cellular viability of tumor chemoresistant high-grade serous carcinoma and non-tumoral cell lines.** (**A**) The dose–response curves for OVCAR8, OVCAR8 PTX R C, and HOSE6.3 cells were obtained by PB assay after exposure to increasing concentrations of GAL (7.81 to 1000 µM). (**B**) Representative microscopy images of OVCAR8, OVCAR8 PTX R C, and HOSE6.3 cells after exposure to increasing concentrations of GAL (0 to 1000 µM) for 48 h. All the figures were taken at 50× magnification, and the scale bar corresponds to 200 μm. All assays were performed in triplicate in at least three independent experiments. Data are expressed as mean ± standard deviation and plotted using GraphPad Prism Software Inc. V8. Statistical analysis was performed using ordinary two-way ANOVA followed by Šidák multiple comparison test. GAL, galanthamine; PB, Presto Blue.

**Figure 3 biomedicines-13-02852-f003:**
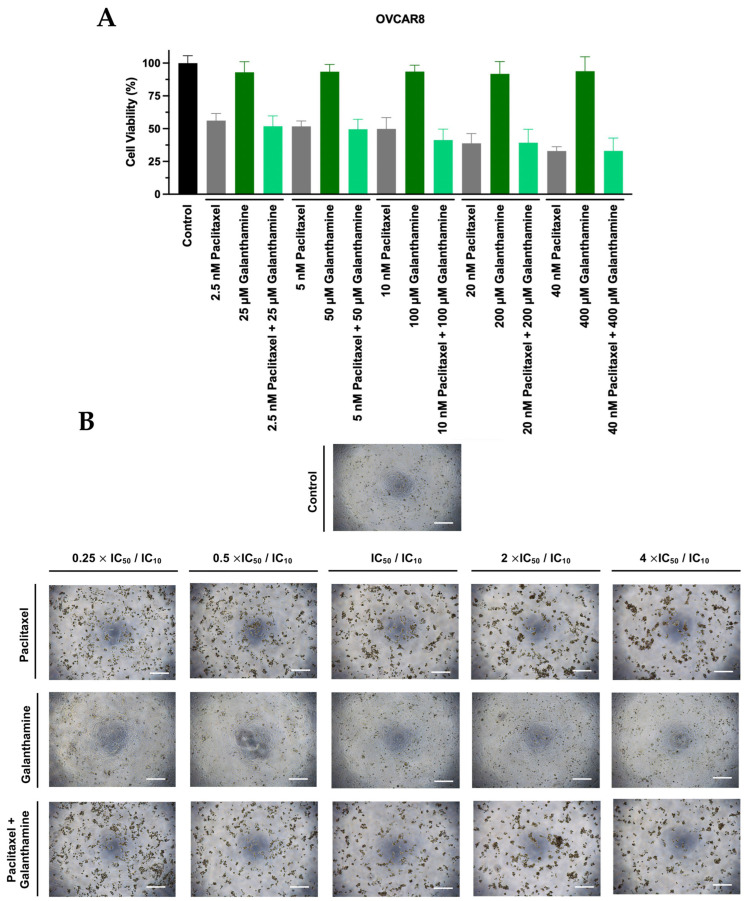
**Galanthamine does not affect the efficacy of paclitaxel in reducing the cellular viability of OVCAR8.** (**A**) Bar charts represent the cellular viability of OVCAR8 cells obtained by PB assay after exposure to a fixed-dose ratio that corresponds to 0.25, 0.5, 1, 2, and 4 times the individual IC_50_ and IC_10_ values of PTX and GAL, respectively, for 48 h. (**B**) Representative microscopy images of OVCAR8 cells after exposure to the control, PTX, GAL, and PTX + GAL at a concentration of 0.25, 0.5, 1, 2, and 4 times the individual IC_50_ and IC_10_ values of PTX and GAL, respectively, for 48 h. All the figures were taken at 50× magnification, and the scale bar corresponds to 200 μm. The combined treatment was administered simultaneously. All assays were conducted in triplicate across at least three independent experiments. Data are presented as mean ± standard deviation and visualized with GraphPad Prism Software Inc. v8. Statistical analysis involved an ordinary one-way ANOVA, followed by Šidák’s multiple comparison test. GAL, galanthamine; PB, Presto Blue; PTX, paclitaxel.

**Figure 4 biomedicines-13-02852-f004:**
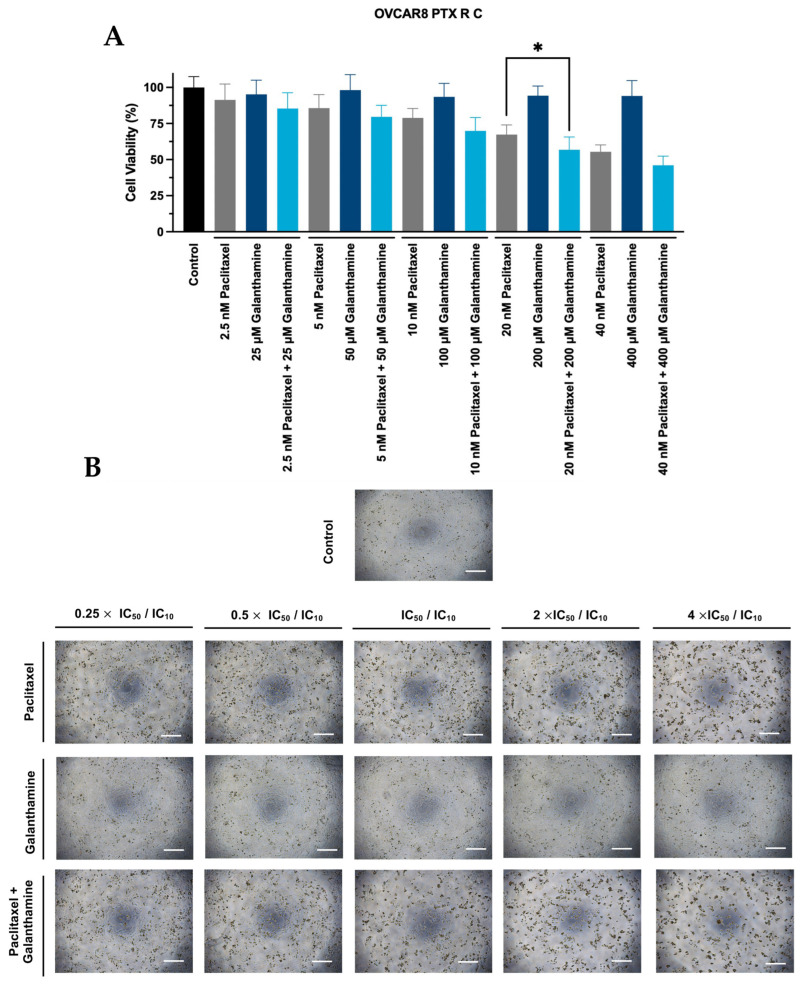
**Galanthamine does not affect the efficacy of paclitaxel in reducing the cellular viability of OVCAR8 PTX R C** (**A**) Bar charts represent the cellular viability of OVCAR8 PTX R C cells obtained by PB assay after exposure to a fixed-dose ratio that corresponds to 0.25, 0.5, 1, 2 and 4 times the individual IC_50_ and IC_10_ values of PTX and GAL, respectively, for 48 h. (**B**) Representative microscopy images of OVCAR8 PTX R C cells after exposure to control, PTX, GAL, and PTX + GAL at a concentration of 0.25, 0.5, 1, 2, and 4 times the individual IC_50_ and IC_10_ values of PTX and GAL, respectively, for 48 h. All the figures were taken at 50× magnification, and the scale bar corresponds to 200 μm. The combined treatment was administered simultaneously. All assays were conducted in triplicate across at least three independent experiments. Data are presented as mean ± standard deviation and visualized with GraphPad Prism Software Inc. v8. Statistical analysis involved an ordinary one-way ANOVA, followed by Šidák’s multiple comparison test, and a *p*-value of * < 0.05 was considered statistically significant. GAL, galanthamine; PB, Presto Blue; PTX, paclitaxel.

**Figure 5 biomedicines-13-02852-f005:**
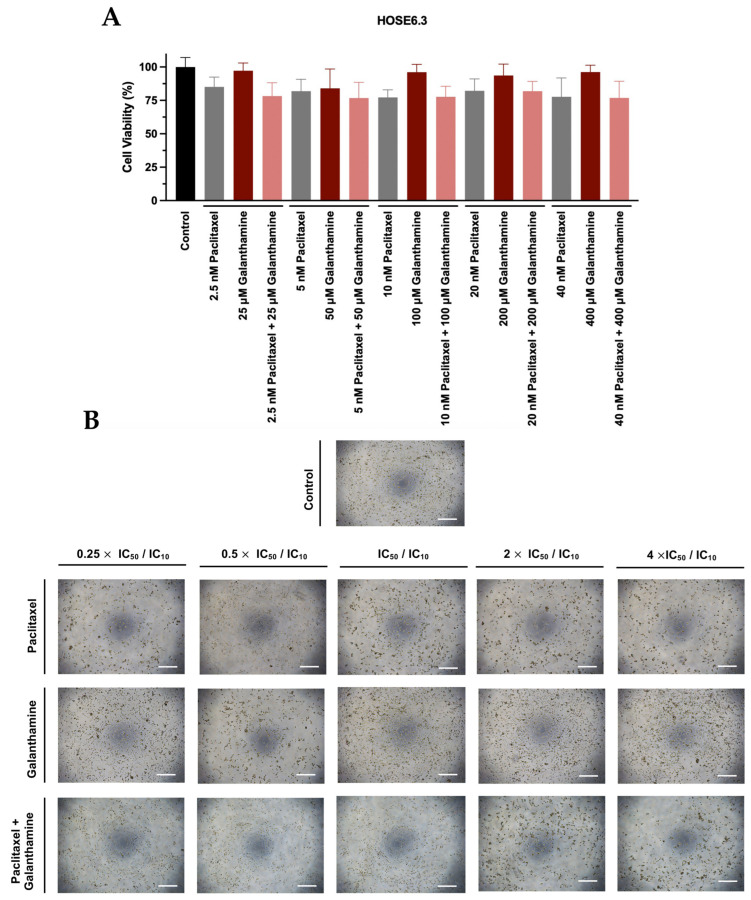
**Galanthamine does not affect the efficacy of paclitaxel in reducing the cellular viability of HOSE6.3.** (**A**) Bar charts represent the cellular viability of HOSE6.3 cells obtained by PB assay after exposure to a fixed-dose ratio that corresponds to 0.25, 0.5, 1, 2, and 4 times the individual IC_50_ and IC_10_ values of PTX and GAL, respectively, for 48 h. (**B**) Representative microscopy images of HOSE6.3 cells after exposure to the control, PTX, GAL, and PTX + GAL at a concentration of 0.25, 0.5, 1, 2, and 4 times the individual IC_50_ and IC_10_ values of PTX and GAL, respectively, for 48 h. All the figures were taken at 50× magnification, and the scale bar corresponds to 200 μm. The combined treatment was administered simultaneously. All assays were conducted in triplicate across at least three independent experiments. Data are presented as mean ± standard deviation and visualized with GraphPad Prism Software Inc. v8. Statistical analysis involved an ordinary one-way ANOVA, followed by Šidák’s multiple comparison test GAL, galanthamine; PB, Presto Blue; PTX, paclitaxel.

**Figure 6 biomedicines-13-02852-f006:**
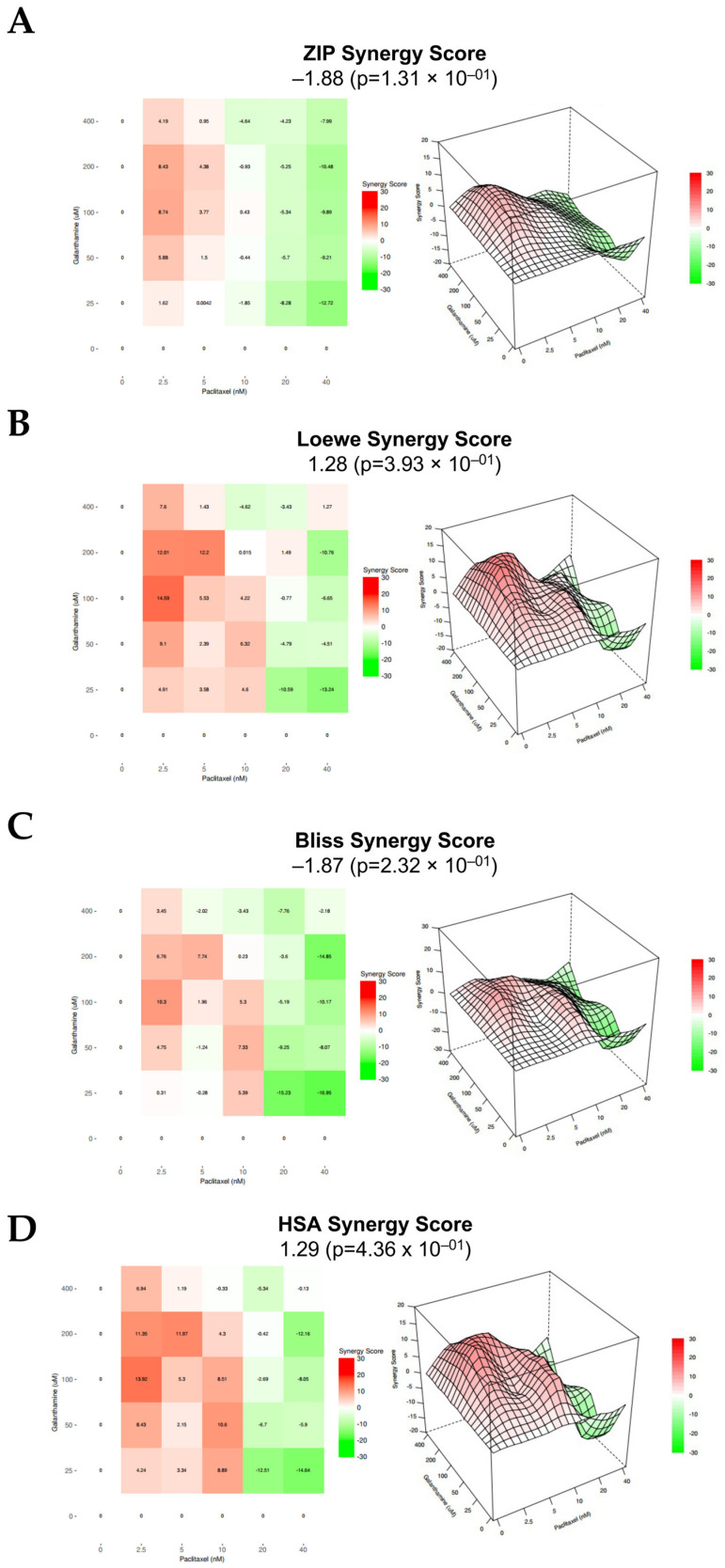
**Combining paclitaxel with galanthamine has an additive effect on OVCAR8.** (**A**–**D**) ZIP, Loewe, Bliss Independence, and HSA reference models. Heat map and 3D plots showing the addictive effect of OVCAR8 cells after exposure to a fixed-dose ratio that corresponds to 0.25, 0.5, 1, 2, and 4 times the individual IC_50_ and IC_10_ values of PTX and GAL, respectively, for 48 h. The combined treatment was administered simultaneously. All assays were performed in triplicate in at least three independent experiments. Synergy score: <−10 (antagonism, green), −10 to 10 (additivity, white), and >10 (synergism, red). GAL, galanthamine; HSA, high single agent; PTX, paclitaxel; ZIP, zero interaction potency.

**Figure 7 biomedicines-13-02852-f007:**
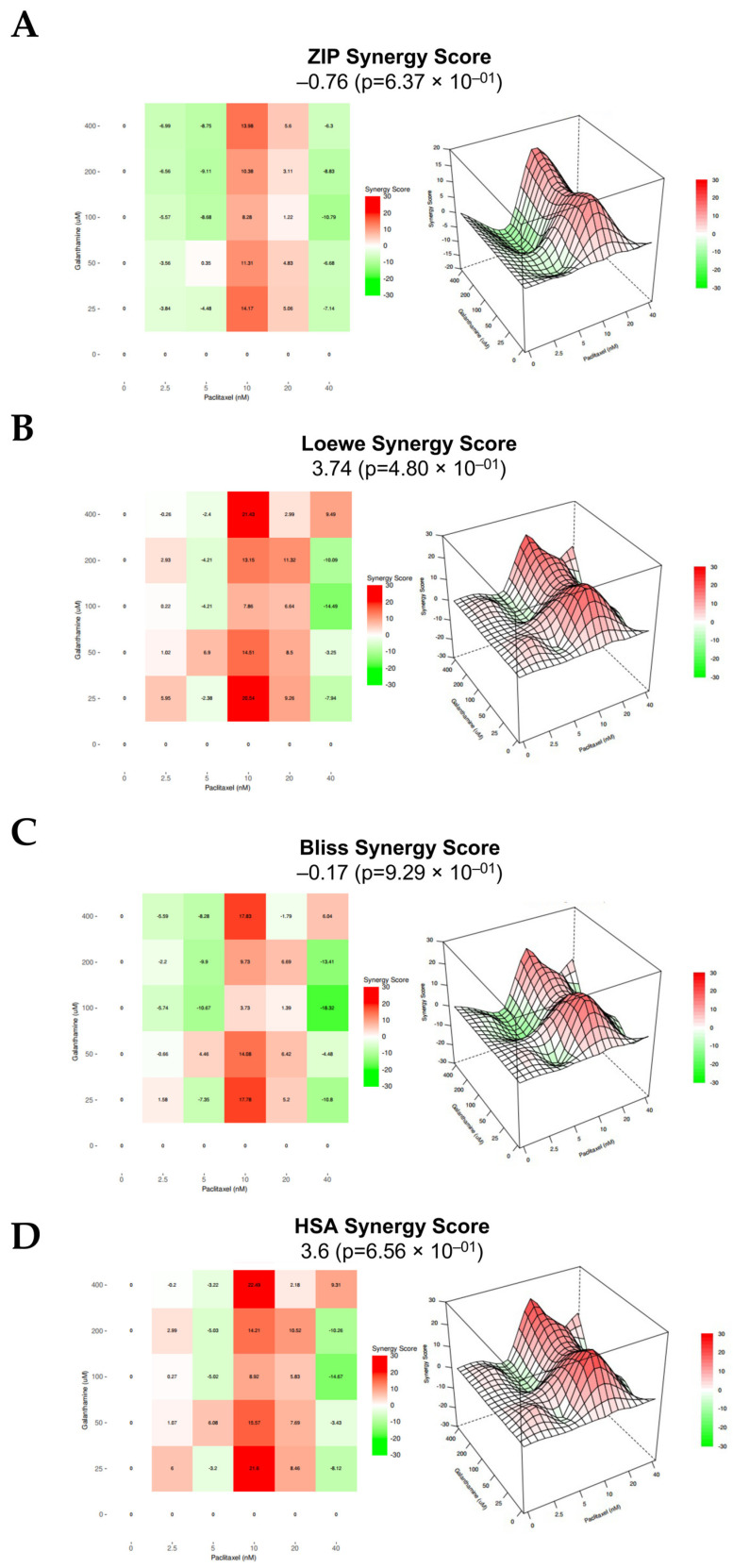
**Combining paclitaxel with galanthamine has an additive effect on OVCAR8 PTX R C.** (**A**–**D**) ZIP, Loewe, Bliss Independence, and HSA reference models. Heat map and 3D plots showing the addictive effect of OVCAR8 PTX R C cells after exposure to a fixed-dose ratio that corresponds to 0.25, 0.5, 1, 2, and 4 times the individual IC_50_ and IC_10_ values of PTX and GAL, respectively, for 48 h. The combined treatment was administered simultaneously. All assays were performed in triplicate in at least three independent experiments. Synergy score: <−10 (antagonism, green), −10 to 10 (additivity, white), and >10 (synergism, red). GAL, galanthamine; HSA, high single agent; PTX, paclitaxel; ZIP, zero interaction potency.

**Figure 8 biomedicines-13-02852-f008:**
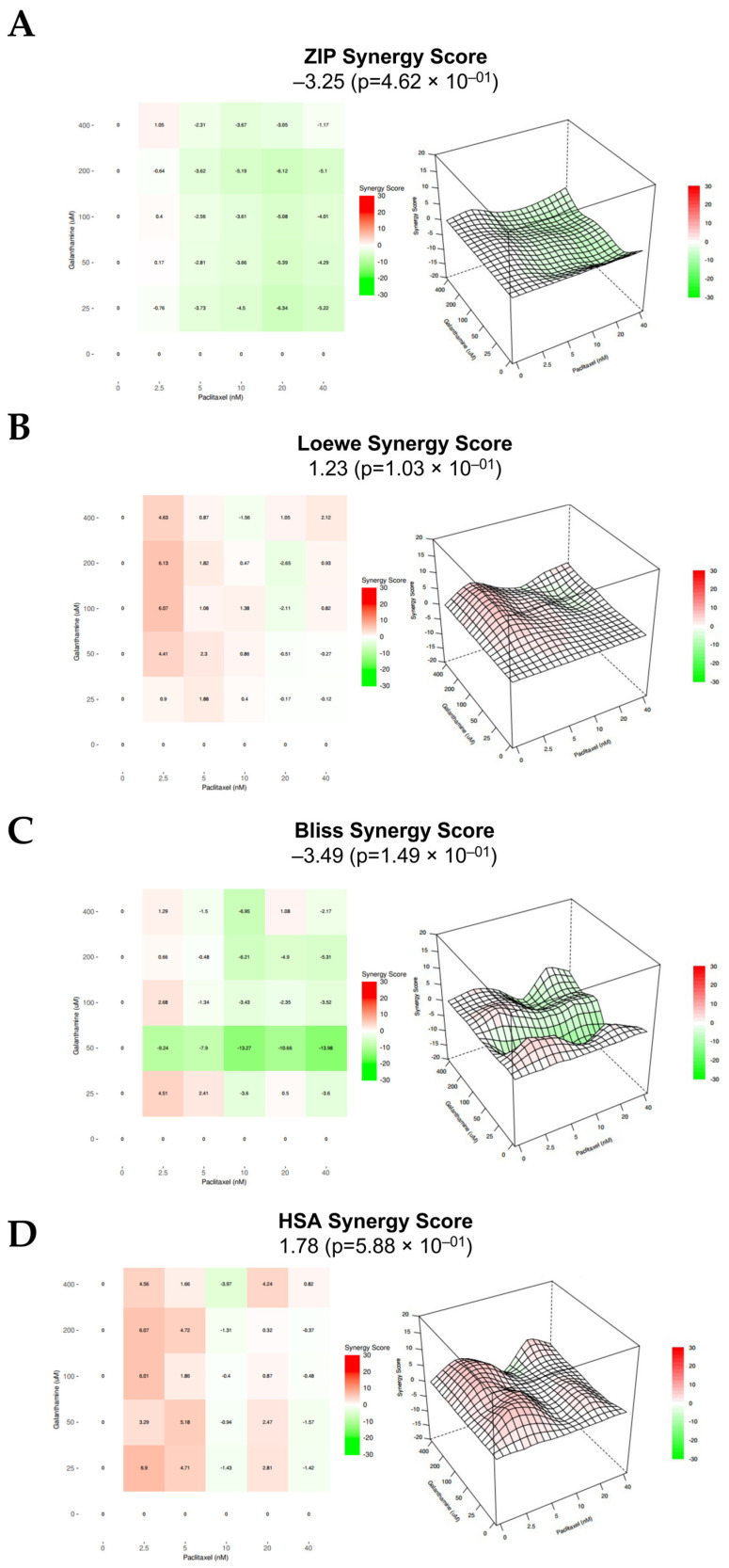
**Combining paclitaxel with galanthamine has an additive effect on HOSE6.3.** (**A**–**D**) ZIP, Loewe, Bliss Independence, and HSA reference models. Heat map and 3D plots showing the addictive effect of HOSE6.3 cells after exposure to a fixed-dose ratio that corresponds to 0.25, 0.5, 1, 2, and 4 times the individual IC_50_ and IC_10_ values of PTX and GAL, respectively, for 48 h. The combined treatment was administered simultaneously. All assays were performed in triplicate in at least three independent experiments. Synergy score: <−10 (antagonism, green), −10 to 10 (additivity, white), and >10 (synergism, red). GAL, galanthamine; HSA, high single agent; PTX, paclitaxel; ZIP, zero interaction potency.

**Figure 9 biomedicines-13-02852-f009:**
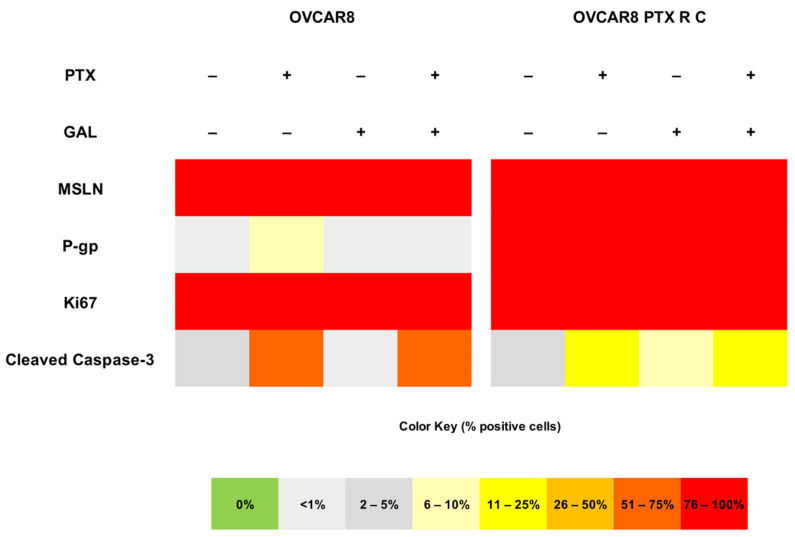
**Galanthamine does not influence the phenotype of both high-grade serous carcinoma cell lines.** The heat map illustrates the percentage of positive cells for mesothelin, P-glycoprotein, Ki-67, and cleaved caspase-3 in both OVCAR8 and OVCAR8 PTX R C cell lines treated with paclitaxel and galanthamine alone and in combination. The color key represents the percentage of positive cells for each marker. GAL, galanthamine; MSLN, mesothelin; P-gp, P-glycoprotein; PTX, paclitaxel.

**Figure 10 biomedicines-13-02852-f010:**
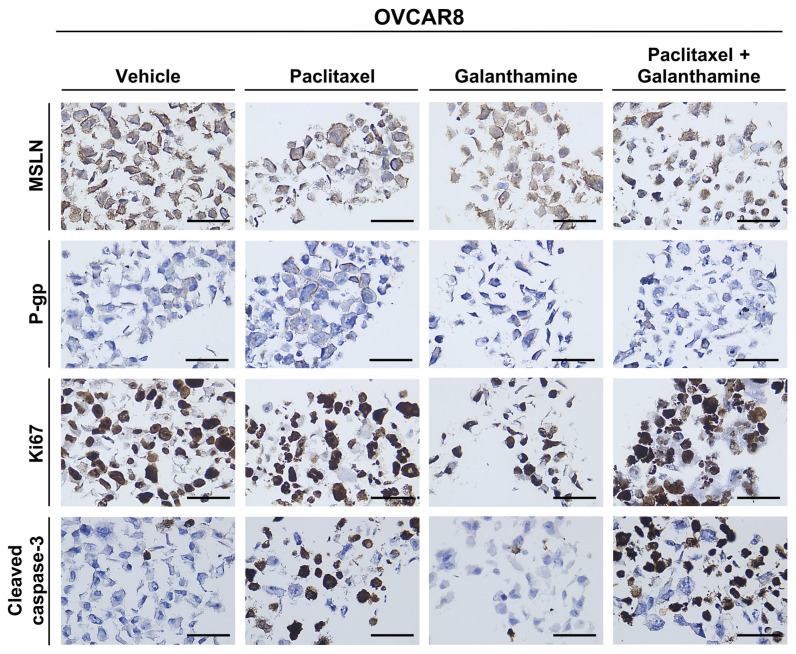
**Galanthamine does not influence the phenotype of OVCAR8**. Representative immunocytochemistry images for mesothelin, P-glycoprotein, Ki67, and cleaved caspase-3 markers in OVCAR8 cells treated with paclitaxel and galanthamine alone and in combination. All the figures were taken at 400× magnification, and the scale bar corresponds to 50 μm. MSLN, mesothelin; P-gp, P-glycoprotein.

**Figure 11 biomedicines-13-02852-f011:**
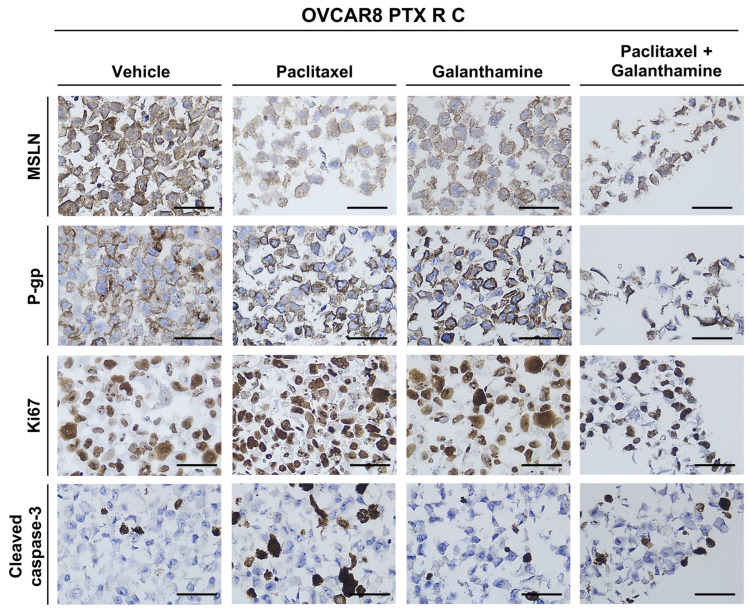
**Galanthamine does not influence the phenotype of OVCAR8 PTX R C.** Representative immunocytochemistry images for mesothelin, P-glycoprotein, Ki67, and cleaved caspase-3 markers in OVCAR8 PTX R C cells treated with paclitaxel and galanthamine alone and in combination. All the figures were taken at 400× magnification, and the scale bar corresponds to 50 μm. MSLN, mesothelin; P-gp, P-glycoprotein.

**Figure 12 biomedicines-13-02852-f012:**
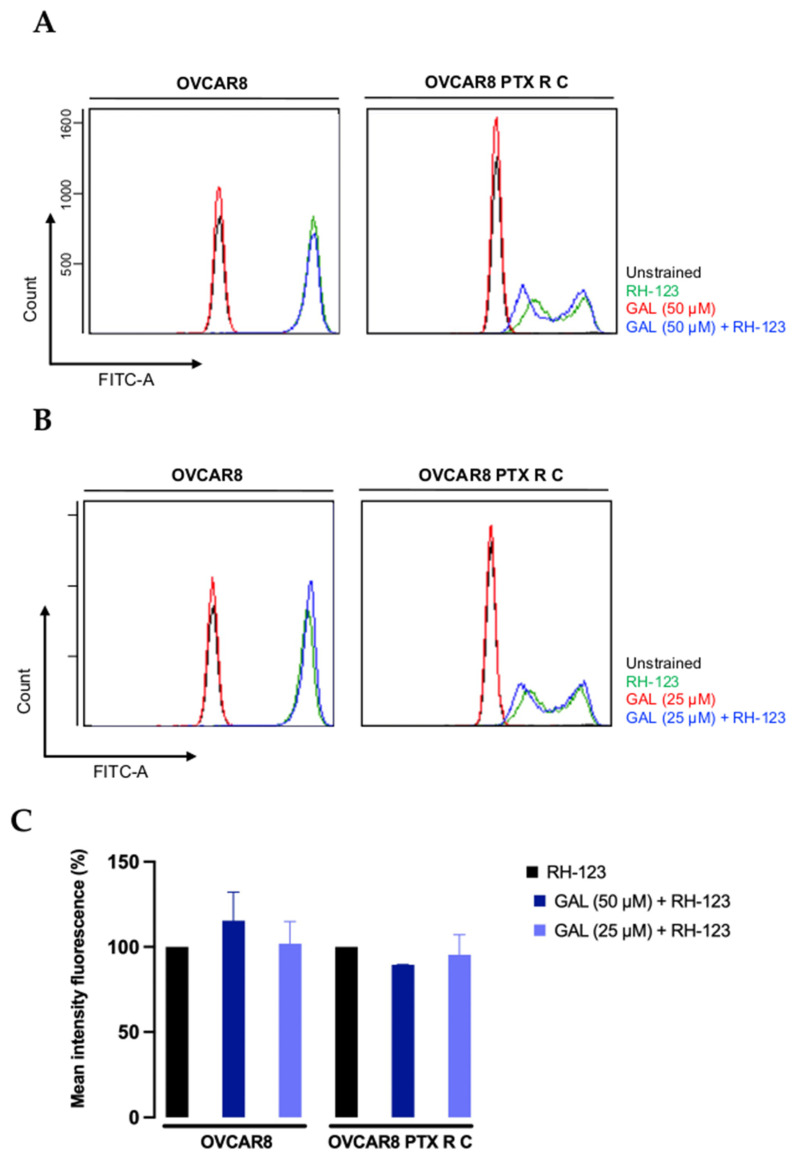
**Rhodamine-123 accumulation after galanthamine treatment of chemoresistant cell lines.** Representative flow cytometry histogram demonstrating RH-123 accumulation using FITC-A intensity in untreated (unstained, black) and RH-123-treated (stained, green) cells in the presence of GAL (unstained, red; stained, blue) at 50 µM (**A**) and 25 µM (**B**) for OVCAR8 and OVCAR8 PTX R C cells. (**C**) Representative flow cytometry bar chart demonstrating the percentage of mean intensity fluorescence, indicating the intracellular accumulation of RH-123 for OVCAR8 and OVCAR8 PTX R C cells. All assays were performed in triplicate in at least three independent experiments. Data are expressed as mean ± standard deviation and plotted using GraphPad Prism Software Inc. v8. Statistical analysis was performed using ordinary one-way ANOVA followed by Šidák’s multiple comparison test. GAL, galanthamine; PTX, paclitaxel.

**Table 1 biomedicines-13-02852-t001:** Synergy score and most synergic area score values obtained for OVCAR8, OVCAR8 PTX R C, and HOSE6.3 cell lines, after exposure to a fixed-dose ratio that corresponds to 0.25, 0.5, 1, 2, and 4 times the individual IC_50_ and IC_10_ values for paclitaxel and galanthamine, respectively. The combined treatment was administered simultaneously. All assays were performed in triplicate in at least three independent experiments. Synergy score: <−10 (antagonism), −10 to 10 (additivity), and >10 (synergism).

Reference Models		Cell Lines		
		OVCAR8	OVCAR8 PTX R C	HOSE6.3
Zero Interaction Potency	Synergy Score	−1.881	−0.760	−3.248
	Most Synergic Area Score	8.739	14.173	1.053
	Interaction	Additive	Additive	Additive
Loewe	Synergy Score	1.275	3.739	1.228
	Most Synergic Area Score	14.587	21.429	6.130
	Interaction	Additive	Additive	Additive
Bliss Independence	Synergy Score	−1.868	−0.171	−3.491
	Most Synergic Area Score	10.303	17.833	4.507
	Interaction	Additive	Additive	Additive
Highest Single Agent	Synergy Score	1.290	3.601	1.784
	Most Synergic Area Score	13.919	22.486	6.900
	Interaction	Additive	Additive	Additive

## Data Availability

The original contributions presented in this study are included in the article. Further inquiries can be directed to the corresponding authors.

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
