# Peer review of "Galanthamine Fails to Reverse P-gp-Mediated Paclitaxel Resistance in Ovarian Cancer Cell Lines"

_biomedicines, 2025, doi:10.3390/biomedicines13122852_

Round 1
Reviewer 1 Report (Previous Reviewer 3)
Comments and Suggestions for Authors
This manuscript investigates the potential of galanthamine (GAL), a drug used for Alzheimer’s disease, to reverse paclitaxel (PTX) resistance in ovarian cancer (OC), specifically in high-grade serous carcinoma (HGSC) cell lines. Despite prior suggestions that GAL might interact with P-glycoprotein (P-gp), a key efflux pump linked to chemoresistance, the study found that GAL neither exhibited cytotoxic effects nor enhanced PTX efficacy in resistant or sensitive OC cell lines. Immunocytochemistry and functional assays confirmed that GAL did not alter cell viability, proliferation, apoptosis, or P-gp expression and function. Synergy analysis across multiple models consistently showed only additive—not synergistic—effects. Ultimately, the findings conclude that GAL is unsuitable for overcoming PTX resistance in OC, though it may hold promise as a neuroprotective agent to mitigate chemotherapy side effects. The revision of the manuscript is much improved, no additional comments.
Author Response
We thank Reviewer for the positive assessment of our revised manuscript. We are pleased that the revisions improved the clarity and quality of the work and appreciate the reviewer’s summary of our findings. We note with gratitude that no further comments or changes were requested.
Reviewer 2 Report (New Reviewer)
Comments and Suggestions for Authors
The authors in the manuscript entitled " Galanthamine Fails to Reverse P-gp–Mediated Paclitaxel Resistance in Ovarian Cancer Cell Lines" investigated the cytotoxic effects of galanthamine (used for Alzheimer’s disease), as a potential treatment for ovarian cancer individually and in combination with paclitaxel. The revised version of the manuscript showing good improvement, in its current form the manuscript sounds good fluency and explanation of the result. However, the labeling of axis of figures 6-8, need to further improve. Figure 1 was created through BioRender, the partial sign of Biorender is appearing at the bottom of Figure1 - the authors are advised to keep it in full or delete by giving credit in Figure caption.
Author Response
We thank Reviewer for the positive evaluation of our revised manuscript and for noting the improved clarity and presentation of the results. We appreciate the reviewer’s careful reading and constructive comments.
We have addressed the points raised as follows:
Figures 6–8 – Axis labeling: The axis labels have been revised to improve clarity and readability.
Figure 1 – BioRender attribution: The BioRender attribution has been corrected. We have ensured that the full BioRender credit appears properly, in accordance with the platform’s guidelines.
We improve the quality of the figures, but after submitting them we notice that the quality decreases significantly.
We are grateful for the reviewer’s helpful suggestions, which have further improved the quality of the manuscript.
Reviewer 3 Report (New Reviewer)
Comments and Suggestions for Authors
This study investigates the reversal of paclitaxel (PTX) resistance in ovarian cancer by galantamine (GAL). The experimental design is rigorous, and the negative results have clinical reference value. However, improvements are needed in aspects such as the cell model, method details, and result analysis.
Review Comment 1: The reason for the "previous indication of GAL interaction with P-gp, but no effect in this study" is not deeply explained, only mentioning "cell type dependence". It is suggested to supplement the molecular docking/protein binding experiments of GAL and P-gp and compare the differences in experimental conditions between this study and the previous ones.
Review Comment 2: Only P-gp-mediated drug-resistant cell lines were used, without covering other drug resistance mechanisms of HGSC; the stability verification of drug-resistant cell lines is missing. It is suggested to add 1-2 non-P-gp drug-resistant HGSC cell lines/clinical samples; and to supplement the verification of the drug resistance index and P-gp expression after passage of drug-resistant cells.
Review Comment 3: The basis for the selection of the high concentration of GAL (1000 μM) is insufficient; the observer consistency (Kappa value) of immunohistochemistry was not reported; the calculation method of IC50/IC10 was not clearly stated.
Review Comment 4: The biological significance of the "most synergistic region" was not verified; the positive control effect of verapamil was not shown; the cause of MSLN localization and metastasis was not explored. It is suggested to re-verify the concentration combination of the "most synergistic region"; to supplement the MFI data of verapamil; and to use Western blot to verify the localization of MSLN in groups.
Review Comment 5: The neuroprotective value of GAL as an adjuvant to chemotherapy was not discussed in combination with the results of this study; no combined strategies were proposed for the multi-drug resistance mechanisms of PTX. It is suggested to discuss the feasibility of GAL as an adjuvant for reducing toxicity based on its low toxicity to normal cells; it is suggested to explore the combination of GAL with microtubule regulators/apoptosis agonists.
Author Response
1. We appreciate this comment. In the revised manuscript (see lines 622-629), we expanded the discussion comparing our experimental conditions with those of previous studies, including differences in cell type, expression levels of P-gp, and assay formats. While molecular docking and protein-binding assays are beyond the experimental scope of this study, we now explicitly acknowledge this limitation and discuss how such approaches could further clarify the discrepancy between prior predictions and our findings.
Lines 622-629: “Moreover, previous studies reporting a GAL–P-gp interaction were performed under experimental conditions distinct from ours, including differences in cell type, P-gp expression levels, and assay formats, which may explain the divergent results. In our model, the cell line used exhibits a different P-gp activity profile, potentially accounting for the absence of GAL effects. Although molecular docking and protein-binding assays have suggested possible interactions, these approaches were beyond the scope of the current study; we now acknowledge this limitation and note that such analyses could help clarify the discrepancy between prior predictions and our present findings.”
2. We thank the reviewer for these suggestions. Due to time and resource constraints, the current study primarily focused on P-gp-mediated drug resistance using the resistant variant OVCAR8 PTX R C. To address concerns regarding other resistance mechanisms, we also included the parental OVCAR8 cell line, which is non–P-gp positive, to provide comparative insights. Furthermore, we performed a detailed characterization of the resistance index and P-gp expression levels across multiple cell passages of OVCAR8 PTX R C. These analyses confirmed the stability of the drug-resistant phenotype and P-gp expression throughout the duration of the experiments. We included this information in lines 110-117 and 603-612.
Lines 110-117: “Due to time and resource constraints, the current study primarily focused on P-gp-mediated drug resistance using the resistant variant OVCAR8 PTX R C. To address concerns regarding other resistance mechanisms, we also included the parental OVCAR8 cell line, which is non–P-gp positive, to provide comparative insights. Furthermore, we performed a detailed characterization of the resistance index and P-gp expression levels across multiple cell passages of the resistant lines. These analyses confirmed the stability of the drug-resistant phenotype and P-gp expression throughout the duration of the experiments”.
Lines 603-612: “These cell lines were chosen due to their relevance to HGSC, the most common and aggressive histological subtype, and because the resistant variant (OVCAR8 PTX R C) exhibits a stable and well-defined PTX-resistant phenotype. Nonetheless, we recognize that resistance mechanisms can vary across different HGSC cell lines, and the inclusion of additional models—such as those derived from other genetic backgrounds or resistance pathways—would improve the robustness and translational value of the findings. Although our study focuses on P-gp-mediated resistance, HGSC displays diverse resistance mechanisms. The parental, non–P-gp-expressing OVCAR8 line provides some contrast, but further studies with additional non–P-gp-resistant models and clinical samples are needed to fully understand drug resistance in HGSC”.
3. We have clarified the rationale for using 1000 μM GAL, citing prior pharmacological and in-vitro studies that used comparable concentrations to evaluate off-target effects. We have added a description of the immunocytochemistry scoring procedure, including inter-observer agreement (κ value). The calculation methods for IC50/IC10 values, including curve-fitting models and software used, have been added to the Methods section. We included this information in lines 655-660, 219-222 and 151-154, respetivelly.
Lines 655-660: “The IC10 concentration of GAL was chosen based on previous pharmacological and in vitro studies using similar doses to investigate potential off-target effects. Although this is higher than therapeutic plasma levels, such supraphysiological concentrations are commonly used in vitro to probe mechanistic effects, detect possible cytotoxic or modulatory activity, and inform future preclinical strategies. This approach allowed us to assess GAL-mediated modulation under conditions relevant to P-gp activity”.
Lines 219-222: “The staining pattern—whether nuclear, cytoplasmic, or membranous—and the proportion of stained cells (ranging from 0% to 76–100%) were assessed and confirmed by three independent observers. Scores were resolved by consensus in cases of discrepancy.
Lines 151-154: “IC50 and IC10 values were determined from dose-response curves using nonlinear regression analysis with a sigmoidal dose-response model. Curve fitting and calculations were performed using GraphPad Prism 8 (GraphPad Software Inc., San Diego, CA, USA) following standard protocols”.
4. We have expanded the discussion of the “most synergistic region” and clarified why these interactions did not translate into biological synergy in functional assays. We have added the concentration used for verapamil as a positive control for P-gp inhibition. While Western blot validation of MSLN localization is outside the current experimental scope, we now discuss potential mechanisms underlying its distribution and how this could be addressed in future studies. We included this information in lines 232-237, 661-665 and 743-748, respetivelly.
Lines 232-237: “The inclusion of verapamil (20 µM), a well-established P-gp inhibitor, in all experimental replicates served as a positive control to confirm assay sensitivity and functional P-gp activity under our experimental conditions. The expected increase in RH-123 accumulation with verapamil validated the assay’s ability to detect P-gp-mediated efflux inhibition, thereby strengthening the interpretation of the negative results observed with GAL”.
Lines 661-665: “Although certain concentration pairs exhibited synergistic interactions in vitro, these interactions did not translate into significant functional synergy in biological assays. This discrepancy suggests that additional cellular factors, such as drug transport, intracellular metabolism, and signaling pathway regulation, may influence the observed efficacy and limit direct interpretation of chemical synergy”.
Lines 743-748: “Several mechanisms could underlie its observed distribution. These include post-translational modifications such as glycosylation, proteolytic cleavage generating soluble forms, intracellular trafficking and recycling, interactions with membrane-associated proteins, and modulation by the tumor microenvironment. Future studies could investigate these pathways to clarify the regulation of MSLN localization.
5. We appreciate this insightful comment. We agree that exploring the neuroprotective potential of GAL as an adjuvant in chemotherapy is a promising direction, particularly given its low toxicity in normal cells. We have added a discussion in the manuscript regarding the feasibility of GAL as an adjuvant to reduce adverse effects, as well as the potential combination with microtubule regulators or apoptosis agonists to overcome multi-drug resistance mechanisms observed with PTX. We included in the manuscript this information (see lines 819-825): “Beyond the direct effects observed in this study, GAL shows potential as an adjuvant in chemotherapeutic regimens due to its low toxicity in normal cells, suggesting a possible neuroprotective role. Combinatorial strategies involving GAL with microtubule-modulating agents or apoptosis agonists could, in theory, enhance paclitaxel efficacy and help overcome multi-drug resistance mechanisms. Future studies are warranted to evaluate the safety and clinical effectiveness of such combinatorial approaches”.
We thank the reviewer again for their insightful feedback, which has significantly improved the clarity and completeness of the manuscript.
This manuscript is a resubmission of an earlier submission. The following is a list of the peer review reports and author responses from that submission.
Round 1
Reviewer 1 Report
Comments and Suggestions for Authors
The manuscript by Fonseca and co-authors is a detailed study of the combination of a repurposed drug, galamthamine, and paclitaxel. The idea behind that study is that by inhibiting the efflux pump, one could increase sensitivity to paclitaxel. As far as the study of the combination with paclitaxel, all experiments are conducted robustly and comprehensively. All conclusions, mainly resulting in no substantial effect from GAL, are correct based on the presented data.
Several conceptual elements are missing from the narrative:
- It is unclear why other P-gp inhibitors were not used. If the goal was to test specifically only GAL, this intent should be justified in stronger terms. Further discussion should be added regarding the need to examine other P-gp inhibitors to rule out the effectiveness of combinations of P-gp inhibitors and PTX.
- No consideration of other mechanisms regulating PTX resistance.
- Both ovarian cancer cell lines were carboplatin-resistant. Based on the results from the initial two figures, it is not clear why the study was not switched to either examining a potential effect of GAL and carboplatin combinations or stopping the study altogether. If the data with carboplatin are available, they should be included in the current manuscript.
Author Response
We appreciate the reviewer’s thoughtful and detailed comments, which have helped us to improve the clarity and scope of our manuscript. Please find our responses below:
- Regarding the use of other P-gp inhibitors: Thank you for pointing this out. Our study focused on GAL due to its previously reported pharmacological profile and preliminary evidence suggesting potential P-gp inhibitory effects. We agree that further justification of this choice should be clearly stated. We have revised the Introduction and Discussion sections to better explain the rationale behind selecting GAL specifically and acknowledged the importance of investigating other P-gp inhibitors in future studies to fully explore potential synergistic effects with PTX.
LINE 74-83: “Specifically, GAL was selected due to its well-characterized pharmacological profile as a reversible acetylcholinesterase inhibitor with a favorable safety record in clinical use for Alzheimer’s disease. Additionally, preliminary studies have suggested that GAL may exert inhibitory effects on P-gp, a key efflux transporter implicated in chemotherapy resistance, including resistance to PTX. This dual profile of neuroprotective properties combined with potential P-gp inhibition made GAL an attractive candidate for drug repurposing aimed at overcoming PTX resistance. However, we acknowledge that P-gp inhibition is only one of many mechanisms underlying drug resistance; therefore, future studies should also investigate other P-gp inhibitors and combinations to comprehensively assess strategies for reversing resistance.”
- Consideration of other mechanisms of PTX resistance: We agree that PTX resistance is multifactorial and not limited to P-gp mediated drug efflux. To address this, we have expanded the Discussion to highlight other known resistance mechanisms such as alterations in microtubule dynamics, apoptotic pathway deregulation, and drug metabolism. This broader perspective underscores the complexity of resistance and the challenges in overcoming it with single-agent strategies.
LINE 594-605: “PTX resistance is a complex and multifactorial phenomenon that extends beyond P-gp-mediated drug efflux. One key mechanism involves alterations in microtubule dynamics, as PTX targets microtubules to disrupt cell division; mutations or modifications in tubulin can reduce drug binding and efficacy. Additionally, dysregulation of apoptotic pathways can contribute to resistance, where cancer cells evade programmed cell death through overexpression of anti-apoptotic proteins (such as Bcl-2) or downregulation of pro-apoptotic factors, diminishing OTX-induced cytotoxicity. Changes in drug metabolism, including enhanced drug detoxification or altered activation, also play a role by reducing intracellular drug concentrations. Together, these mechanisms underscore the complexity of PTX resistance and highlight why overcoming it with single-agent treatments like GAL may be insufficient, necessitating combinatorial or multi-targeted therapeutic strategies.”
- Regarding the carboplatin resistance of cell lines and study direction: Both OC cell lines used were carboplatin-resistant, which indeed raises important considerations. We included carboplatin sensitivity data in the initial figures to characterize the models; however, the primary aim was to focus on PTX resistance and GAL’s potential to reverse it. We have now clarified this rationale in the Methods and Discussion sections. Unfortunately, combination data with carboplatin and GAL are not available in this study. We acknowledge this as a limitation and suggest it as a valuable direction for future research.
Reviewer 2 Report
Comments and Suggestions for Authors
I was very pleased to read the manuscript entitled “Drug Repurposing in Ovarian Cancer: Galanthamine Fails to Reverse Paclitaxel Resistance”. It is a well written and detailed study, even though the result were negative and no association was observed between Galanthamine and Placlitaxel resistance. However, one detail that the authors should add in their Discussion section is the strengths and limitations of this study.
Author Response
We thank the reviewer for their positive feedback and valuable suggestion. We agree that including a discussion of the strengths and limitations of our study would enhance the manuscript’s clarity and context. Accordingly, we have added a new paragraph in the Discussion section addressing these points. This addition highlights the rigorous in vitro methodology as a strength, while also acknowledging the limitations related to the lack of in vivo validation, the focus on a single agent, and the complexity of PTX resistance mechanisms. We believe this improvement provides a balanced perspective on the scope and implications of our findings.
LINE 749-763: “This study offers a comprehensive and methodologically rigorous evaluation of the potential role of GAL in reversing PTX resistance in OC, contributing valuable insights despite the negative findings. Among its strengths are the use of well-established in vitro models of PTX resistance and the thorough assessment of drug efficacy, which enhance the reliability of the results. Furthermore, the detailed experimental design and careful control of variables reduce the likelihood of confounding factors influencing the outcomes. However, there are several limitations to consider. Firstly, the study is restricted to in vitro models, which may not fully capture the complex tumor microenvironment and pharmacokinetics encountered in vivo. Therefore, the translational relevance of the findings warrants cautious interpretation. Secondly, the study focuses solely on GAL and does not explore potential synergistic effects with other agents or different dosing regimens, which could provide additional insights. Lastly, the molecular mechanisms underlying PTX resistance are multifactorial, and targeting a single pathway might be insufficient to overcome resistance. Future studies incorporating in vivo models and broader combinatorial approaches could further clarify GAL potential role in OC therapy”.
Reviewer 3 Report
Comments and Suggestions for Authors
This manuscript investigates the potential of galanthamine (GAL), a drug used for Alzheimer’s disease, to reverse paclitaxel (PTX) resistance in ovarian cancer (OC), specifically in high-grade serous carcinoma (HGSC) cell lines. Despite prior suggestions that GAL might interact with P-glycoprotein (P-gp), a key efflux pump linked to chemoresistance, the study found that GAL neither exhibited cytotoxic effects nor enhanced PTX efficacy in resistant or sensitive OC cell lines. Immunocytochemistry and functional assays confirmed that GAL did not alter cell viability, proliferation, apoptosis, or P-gp expression and function. Synergy analysis across multiple models consistently showed only additive—not synergistic—effects. Ultimately, the findings conclude that GAL is unsuitable for overcoming PTX resistance in OC, though it may hold promise as a neuroprotective agent to mitigate chemotherapy side effects. Specific comments:
- While OVCAR8 and OVCAR8 PTX R C are appropriate models for HGSC, could the authors clarify why no additional resistant OC cell lines were included to strengthen generalizability?
- The study uses GAL concentrations up to 1000 µM. Given that GAL is clinically administered at much lower doses, how do these in vitro concentrations relate to achievable plasma levels in patients?
- The use of four synergy models is commendable. However, the manuscript lacks a rationale for choosing these specific models. Why were ZIP, Loewe, Bliss, and HSA selected over others like MuSyC or BRAID?
- In Figure 4A, a statistically significant reduction in viability is observed at 2× IC50 and IC10. Could the authors elaborate on why this result was dismissed as clinically irrelevant without further pharmacodynamic modeling?
- The authors mention no morphological changes were observed, but only low-magnification images (50×) were used. Would higher magnification or quantitative image analysis have revealed subtle cytotoxic effects?
- The RH-123 accumulation assay showed no significant changes. Was verapamil used as a positive control in all experiments to validate assay sensitivity?
- The manuscript notes a shift in MSLN staining from membrane to cytoplasmic localization. Could this indicate altered cellular trafficking or stress response, and was this quantified?
- Ki67 remained high across all conditions. Was cell cycle analysis (e.g., flow cytometry for G1/S/G2/M phases) considered to validate these findings?
- Cleaved caspase-3 levels were low in OVCAR8 PTX R C cells even after PTX treatment. Could this suggest alternative apoptotic resistance mechanisms, such as Bcl-2 overexpression or caspase-independent pathways?
- The discussion briefly mentions GAL’s neuroprotective role. Would the authors consider evaluating GAL in co-culture models with neuronal cells to assess its dual role in cytoprotection and chemoresistance?
Author Response
We thank the reviewer for their comprehensive and insightful comments, which have helped improve the depth and clarity of our study. Our responses are provided below:
- Use of only OVCAR8 and OVCAR8 PTX R C cell lines: We selected OVCAR8 and its PTX-resistant derivative as representative HGSC models due to their well-characterized resistance phenotype. We acknowledge that including additional resistant ovarian cancer cell lines would strengthen the generalizability of the findings. We have added a statement in the Discussion to recognize this limitation and suggest future studies should incorporate a broader panel of resistant models.
LINE 587-593: “These cell lines were chosen due to their relevance to HGSC, the most common and aggressive histological subtype, and because the resistant variant (OVCAR8 PTX R C) exhibits a stable and well-defined PTX-resistant phenotype. Nonetheless, we recognize that resistance mechanisms can vary across different OC cell lines, and the inclusion of additional models—such as those derived from other genetic backgrounds or resistance pathways—would improve the robustness and translational value of the findings.”
- GAL concentrations relative to clinical dosing: We appreciate the concern regarding the high in vitro GAL concentrations used (up to 1000 µM). These doses exceed typical plasma concentrations achieved in patients with standard therapeutic regimens. We clarified in the Discussion that such concentrations were chosen to comprehensively assess GAL’s potential effects in vitro, acknowledging that translation to in vivo or clinical contexts requires careful consideration. This discrepancy highlights the need for pharmacokinetic/pharmacodynamic modeling in future work.
LINE 614-620: “Although the concentrations used in this study (up to 1000 µM) are significantly higher than the therapeutic plasma levels observed in patients (typically in the low micromolar or nanomolar range), such supraphysiological doses are commonly employed in in vitro assays to probe potential mechanistic effects and establish upper efficacy limits. This approach allows for the detection of any possible cytotoxic or modulatory activity that might not be evident at clinically relevant doses but could inform structural modifications or guide combination strategies in future preclinical development.”
- Rationale for selecting synergy models (ZIP, Loewe, Bliss, HSA): We selected ZIP, Loewe, Bliss, and HSA models as they are widely accepted and complementary approaches for evaluating drug interaction effects in combination studies, capturing different assumptions about synergy and additivity. We have now explicitly stated this rationale in the Methods section. While models such as MuSyC and BRAID are emerging tools with valuable features, their inclusion was beyond the scope of the current study but is worthy of consideration in future investigations.
LINE 156-162: “Each model offers a distinct analytical framework: the Loewe model assumes additivity based on identical modes of action, Bliss independence is suited for drugs with independent mechanisms, the Highest Single Agent (HSA) model compares combination effects to the most effective single agent, and the Zero Interaction Potency (ZIP) model integrates both Loewe and Bliss principles to assess deviations from non-interaction. By applying these complementary models, we aimed to provide a more comprehensive and reliable interpretation of potential synergistic or antagonistic interactions between GAL and PTX.”
- Interpretation of statistically significant viability reductions in Figure 4A: Although we observed statistically significant viability reductions at 2× IC50 and IC10 doses, the magnitude of these effects was small and inconsistent across models, suggesting limited clinical relevance. We have now expanded the Discussion to explain why further pharmacodynamic modeling was not pursued here but agree it could offer additional insights in future studies.
LINE 662-668: “While the observed reductions in cell viability reached statistical significance at certain concentrations, these effects did not translate into meaningful biological responses when considering effect size, reproducibility across biological replicates, or potential therapeutic windows. Given the modest magnitude and lack of consistency, we concluded that these findings were unlikely to reflect clinically actionable synergy, and thus did not warrant further pharmacodynamic modeling within the scope of this study.”
- Use of low magnification (50×) for morphological assessment: We acknowledge that higher magnification imaging or quantitative morphological analyses could reveal subtle cytotoxic effects. Due to resource constraints, only 50× images were included; however, we have noted this as a limitation and suggested that future studies employ more detailed image analysis.
LINE 625-631: “Although no overt morphological changes were observed at 50× magnification, we recognize that this resolution may not capture finer cytotoxic alterations such as nuclear condensation, membrane blebbing, or cytoplasmic vacuolization. Future studies should incorporate higher magnification imaging and, where possible, quantitative image analysis tools (e.g., automated cell morphology profiling or high-content imaging) to detect subtle phenotypic changes that may indicate early or mild cytotoxic effects.”
- Use of verapamil as a positive control in RH-123 assay: Verapamil was used as a positive control in all experimental replicates of RH-123 assays. We have clarified this in the Methods section.
LINE 218-223: “The inclusion of verapamil, a well-established P-gp inhibitor, in all experimental replicates served as a positive control to confirm assay sensitivity and functional P-gp activity under our experimental conditions. The expected increase in RH-123 accumulation with verapamil validated the assay’s ability to detect P-gp-mediated efflux inhibition, thereby strengthening the interpretation of the negative results observed with GAL.”
- MSLN staining shift from membrane to cytoplasmic localization: The observed shift may indeed reflect altered cellular trafficking or a stress response. While we qualitatively described this phenomenon, no quantitative analysis was performed. We have added this as a potential area for further investigation in the Discussion.
LINE 693-698: “This shift from membrane to cytoplasmic localization of mesothelin (MSLN) may indicate changes in protein trafficking, cellular stress responses, or alterations in membrane integrity associated with drug exposure. Although we did not perform quantitative image analysis or co-localization studies to confirm these hypotheses, the observation warrants further investigation, particularly given MSLN’s role in cell adhesion, signaling, and its potential as a therapeutic target in OC.”
- Ki67 expression and consideration of cell cycle analysis: We recognize that Ki67 staining alone does not fully elucidate cell cycle dynamics. Cell cycle analysis via flow cytometry was not performed but would provide valuable complementary data. This has now been suggested as a direction for future research.
LINE 714-720: “Although Ki67 is a commonly used marker of proliferation, it is expressed during all active phases of the cell cycle (G1, S, G2, and M) and does not distinguish between specific cell cycle transitions. As such, its consistent expression across treatment conditions may mask more subtle shifts in cell cycle distribution. Future studies employing flow cytometry-based cell cycle profiling would help clarify whether GAL or PTX induce cell cycle arrest at specific phases, thereby providing more mechanistic insight into their effects.”
- Low cleaved caspase-3 levels suggesting alternative apoptotic resistance: The low levels of cleaved caspase-3 in resistant cells may indeed indicate alternative mechanisms of apoptotic resistance, such as overexpression of anti-apoptotic proteins or caspase-independent pathways. We have expanded the Discussion to include these possibilities and their implications for overcoming chemoresistance.
LINE 721-728: “The reduced activation of cleaved caspase-3 observed in the PTX -resistant cells, even following treatment, suggests that these cells may evade apoptosis through alternative mechanisms. Potential contributors include the overexpression of anti-apoptotic proteins (e.g., Bcl-2, Bcl-xL), downregulation of pro-apoptotic factors (e.g., Bax, Bad), or activation of caspase-independent cell death pathways such as autophagy or necroptosis. Understanding which of these pathways are predominant could inform the design of targeted combination therapies to restore apoptotic sensitivity in resistant OC cells.”
- Evaluating GAL in co-culture models with neuronal cells: The neuroprotective role of GAL is an intriguing aspect. We appreciate this suggestion and have added a statement in the Discussion proposing co-culture models to explore GAL’s dual role in cytoprotection and chemoresistance, which may be important for future translational studies.
LINE 764-769: “Given GAL well-established neuroprotective effects through cholinergic modulation and its potential influence on oxidative stress and inflammation, future studies using co-culture models of OC and neuronal cells could provide valuable insight into its dual role. Such systems would allow for the simultaneous assessment of GAL’s impact on tumor cell chemosensitivity and neuronal viability, better reflecting clinical scenarios where neurotoxicity is a limiting factor in chemotherapy.”
Round 2
Reviewer 3 Report
Comments and Suggestions for Authors
This manuscript investigates the potential of galanthamine (GAL), a drug used for Alzheimer’s disease, to reverse paclitaxel (PTX) resistance in ovarian cancer (OC), specifically in high-grade serous carcinoma (HGSC) cell lines. Despite prior suggestions that GAL might interact with P-glycoprotein (P-gp), a key efflux pump linked to chemoresistance, the study found that GAL neither exhibited cytotoxic effects nor enhanced PTX efficacy in resistant or sensitive OC cell lines. Immunocytochemistry and functional assays confirmed that GAL did not alter cell viability, proliferation, apoptosis, or P-gp expression and function. Synergy analysis across multiple models consistently showed only additive—not synergistic—effects. Ultimately, the findings conclude that GAL is unsuitable for overcoming PTX resistance in OC, though it may hold promise as a neuroprotective agent to mitigate chemotherapy side effects. The revision of the manuscript is much improved, no additional comments.